



# The Community Firn Model (CFM) v1.0

C. Max Stevens[1], Vincent Verjans[2], Jessica M.D. Lundin[1,3], Emma C. Kahle[1], Annika N. Horlings[1], Brita I. Horlings[1], and Edwin D. Waddington[1]

[1]Department of Earth and Space Sciences, University of Washington, WA USA
[2]Lancaster Environment Centre, Lancaster University, Lancaster, LA1 4YW, UK
[3]Salesforce, San Francisco, CA

**Correspondence:** C. Max Stevens (maxstev@uw.edu)

**Abstract.**

Models that simulate evolution of polar firn are important for several applications in glaciology, including converting ice-sheet elevation-change measurements to mass change and interpreting climate records in ice cores. We have developed the Community Firn Model (CFM), an open-source, modular model framework designed to simulate numerous physical processes in firn. The modules include firn densification, heat transport, meltwater percolation and refreezing, water-isotope diffusion, and firn-air diffusion. The CFM is designed so that new modules can be added with ease. In this paper, we first describe the CFM and its modules. We then demonstrate the CFM's usefulness in two model applications that utilize two of its novel aspects. The CFM currently has the ability to run any of 13 previously published firn-densification models, and in the first application we compare those models' results when they are forced with regional climate model outputs for Summit, Greenland. The results show that the models do not agree well (spread greater than 10%) when predicting depth-integrated porosity, firn age, or trend in surface-elevation change trend. In the second application, we show that the CFM's coupled firn-air and firn-densification models can simulate noble-gas records from an ice core better than a firn-air model alone.

## 1 Introduction

Snow that falls on an ice sheet transitions to ice through an intermediate stage called firn. Knowledge of the physics of firn densification has several applications in glaciology. Studies of ice-sheet mass balance using altimetry methods require knowing the mass, and mass changes, of the firn to estimate the contribution of the ice sheets to sea-level rise (Shepherd et al., 2012; The IMBIE Team, 2018). Ice-core studies require knowledge of the age of the firn at the depth where bubbles of air become trapped in order to determine the difference between the age of air in the bubbles and the ice that encloses the bubbles (called $\Delta$age; Blunier and Schwander, 2000). Both of these applications require a firn-densification model. In addition, ice-core researchers use firn-air models to simulate the diffusion of atmospheric gases through the porous firn; these models can be used, for example, to estimate the age of gases when they become trapped in bubbles (e.g. Buizert et al., 2012).





Firn is commonly divided by density into three zones based on the dominant physics of densification (Herron and Langway, 1980; Maeno and Ebinuma, 1983). The first zone is defined to include the firn from the surface density (often assumed to
be $\sim 300 - 350 \, \mathrm{kg \, m^{-3}}$ in polar regions) to $550 \, \mathrm{kg \, m^{-3}}$. In zone 1, densification is usually considered to be due to grain boundary sliding and settling (Alley, 1987). In zone 2, which spans the densities between $550 \, \mathrm{kg \, m^{-3}}$ and $\sim 830 \, \mathrm{kg \, m^{-3}}$, densification occurs due to sintering processes (Gow, 1975). Near a density of $830 \, \mathrm{kg \, m^{-3}}$, bubbles of air become trapped and further densification is due to compression of the bubbles. This density is referred to as the bubble-close-off (BCO) density, which is reached at the corresponding BCO depth.

Numerous models have been developed to describe the physics of firn densification. In addition to predicting evolution of firn density, most firn-densification models also simulate the firn's temperature evolution by coupling a heat-diffusion model. A common way to form a firn-densification model is to assume that, for a given site, the accumulation rate is constant and the firn-density profile is in steady state. Using this steady-state assumption, known as Sorge's Law, the change in density $\rho$ with depth $(d\rho/dz)$ can be converted to a material-following (Lagrangian) change in density with time $(d\rho/dt)$ (Bader, 1954). Using
depth-density data from many sites, firn-densification models can be formulated as a function of temperature (often through an Arrhenius term with a tuned activation energy), accumulation rate (a proxy for stress), and one or more tuning parameters.

Several firn-densification models have been developed without invoking Sorge's Law. Some of these have used firn strain-rate data; these include work by Arthern et al. (2010), who measured firn compaction rates in real time in Antarctica using "coffee-can" type strain gauges (Hulbe and Whillans, 1994) and by Morris and Wingham (2014), who inferred firn-compaction
rates by tracking layering in repeated high-resolution density logs of boreholes. Other studies have worked to develop a model based on the microphysical processes driving densification, e.g. Alley (1987) developed a model for zone 1 densification by applying grain-boundary sliding theory. Arnaud et al. (2000) combined the work by Alley (1987) with theory describing pressure-sintering of spherical powders (Arzt, 1982) to simulate densification in zone 2.

The evolution of firn density is governed by grain-scale (i.e. microstructural) processes (Arnaud et al., 2000; Morris and
Wingham, 2014), but most firn models predict density evolution based only upon the accumulation rate and temperature. These fields can come from a regional climate model (RCM), e.g. the Regional Atmospheric Climate Model (RACMO; Noël et al., 2018) or Modèle Atmosphérique Régional (MAR; Fettweis et al., 2017); in-situ weather stations such as the Greenland climate network (GC-Net; Steffen and Box, 2001; Vandecrux et al., 2018), or ice-core data (e.g. Buizert et al., 2015). Firn-densification models also need a surface-density boundary condition; this can be assumed to be constant through time (Fausto et al., 2018),
or it can be predicted, for example, by using an empirical parameterization based on temperature (e.g. Kuipers Munneke et al., 2015) or some other variable.

Atmospheric gases move through the firn's pore space above the BCO depth. Below the BCO depth these gases are trapped in bubbles, and they preserve a record of past atmospheric composition. Gas transport in firn is commonly modeled by dividing the firn into three zones by dominant transport mechanisms, which differ from the three zones of densification (Sowers et al.,
1992). Near the surface is the convective zone, which may be from 0 to $\sim$20 m thick; in this zone, convective mixing due to wind pumping and buoyancy dominates gas transport and keeps the air well-mixed and in equilibrium with the free atmosphere above (Kawamura et al., 2006). Below the convective zone is the diffusive zone, where gas transport is driven primarily



by diffusion along chemical-concentration gradients. Additionally, isotopic fractionation occurs in the diffusive zone due to gravitational and thermal effects: gravitational fractionation causes heavier isotopes to become enriched (i.e. their relative abundance increases) at greater depths, and thermal fractionation causes heavier isotopes to become enriched at the cold end of a temperature gradient (Severinghaus et al., 1998). Numerous models have been developed to simulate air movement in firn; they aid in using firn-air measurements to reconstruct past atmospheric conditions (Buizert et al., 2012).

We have developed the Community Firn Model (CFM), an open-source model framework that includes a suite of published firn-densification models, a firn-air model, and numerous modules to simulate other physical processes in firn. We created the CFM to be a resource to the glaciological community at large. We recognize that many research groups have their own firn models, but our goals with the CFM are (1) to provide a model to research groups who need firn-model outputs but do not want to code a model themselves; (2) to provide a point of reference for research groups to compare their model output against; and (3) to enable firn-densification model comparisons within a single model framework to improve understanding of firn-model uncertainties within various applications. In this paper, we describe the model and demonstrate its utility in two model applications.

## 2 The Community Firn Model

The CFM is an open-source, modular firn-model framework. It is coded in Python 3 and is available for download on GitHub. It is designed to simulate numerous processes associated with firn; "modular" refers to the fact that the CFM was constructed so that the user can choose which of these processes she or he would like to simulate in a given model run, and a module is a piece of code that simulates a particular process (e.g. density evolution). Modules to simulate additional processes can be added with minimal alteration of existing code. The CFM's modularity allows the user to easily choose which physical processes to simulate and which model outputs to save in a particular CFM run. The core modules of the CFM track evolution of the firn density and temperature. Other modules simulate grain-size evolution, firn-air diffusion, water-isotope diffusion, meltwater percolation and refreezing, and layer thinning due to horizontal strain; these latter modules also require execution of the core modules.

### 2.1 CFM workflow

Prior to running the CFM, the user sets the parameters specific to the model run in a .json-formatted configuration file. These include, among others, which firn-densification physics to use, the time-step size, and the model-domain thickness. The configuration file also includes the paths to the files used for forcing the model (i.e. the surface boundary conditions). The CFM is forced at the upper boundary (i.e. the ice-sheet surface) using surface temperature (i.e. the temperature of the snow at the surface), surface density, accumulation-rate data, and any other surface boundary condition needed for a particular module. These fields are input via a .csv file that includes time in the first row and value in the second row.

A CFM run begins by first "spinning up" the model to a steady state, which becomes the initial condition for the "main" model run. During spin up, the CFM uses a steady-state temperature $T_0$ and accumulation rate $\dot{b}_0$, taken as either the mean





or initial value of the forcing data. The CFM first creates depth-density and depth-age profiles using the Herron and Langway (1980) steady-state analytic firn-densification model with $\dot{b}_0$ and $T_0$. It then time steps forward, using $\dot{b}_0$ and $T_0$ and the specified firn-densification model to evolve the firn. The CFM uses a Lagrangian grid with fixed number of model volumes; each volume represents a layer of firn with uniform properties. At each time step, accumulation is added as a new volume at the surface, and a volume is removed from the bottom of the grid. Although the user specifies how long the spin-up should last, it is recommended to spin up long enough to reset the entire grid (i.e. to flush out all of the initial volumes and replace them with new volumes). During spin up, the CFM evolves the density, temperature, age, and other properties that might be included in a model run (e.g. grain size).

After the spin up has completed, the "main" model run begins. It operates in the exact same way as the spin up, except the model is forced with varying temperature, accumulation rate, and other boundary conditions, rather than the steady-state values. When the model run is complete, the model outputs are saved in a single .hdf5-formatted file. The user specifies which model outputs to save; the options are firn depth, density, age, temperature, compaction rate, grain size, water-isotope values, BCO depth and age, depth-integrated porosity, liquid water content, and gas concentrations.

We next describe the various modules built into the CFM, with particular focus on the firn-density and firn-air modules.

## 2.2 Density

The CFM is coded to include 13 previously published firn-densification models (listed in Table 1) and it is designed so that it is easy for the user to choose which firn-densification model to use in a particular CFM run. We note that the word "model" can be ambiguous: we use "CFM" to refer to the entire Community Firn Model framework, and we use "firn-densification model" to refer to an equation or set of equations that simulates the physics of firn densification. Thus, running the CFM includes implementing a firn-densification model.

A general form used in many firn-densification models assumes that a firn layer's change in density $\rho$ through time $t$ is a function of the temperature $T$, accumulation rate $\dot{b}$, and current density:

$$\frac{d\rho}{dt} = f(T, \dot{b}, \rho). \tag{1}$$

Density evolution in the CFM is handled with an explicit numeric scheme, i.e.

$$\rho_{new} = \rho_{old} + (d\rho/dt)dt. \tag{2}$$

Most of the firn-densification models in the CFM use accumulation rate as a proxy for the stress. If accumulation rate is constant in time, the overburden stress $\sigma$ at depth $z$ is related to the mass accumulation rate $\dot{b}$ by the relation $\sigma(z) = \dot{b}g\tau(z)$, where $g$ is gravity and $\tau(z)$ is the age of the firn at depth $z$. For the models that are forced with accumulation rate (as opposed to stress), the CFM by default uses the mean accumulation rate $\bar{\dot{b}}$ over the lifetime of each parcel of firn, rather than the





**Table 1.** List of firn-densification models and their abbreviations coded in the CFM and included in the study detailed in Section 3. Arthern et al. (2010) describe two different models; see Section 2.2.6 for details.

| Model Name/Reference | Abbreviation |
| --- | --- |
| Herron and Langway (1980) | HL |
| Barnola et al. (1991) | BAR |
| Goujon et al. (2003) | GOU |
| Li and Zwally (2011, 2015) | LZ11, LZ15 |
| Helsen et al. (2008) | HEL |
| Arthern et al. (2010) | ART-T, ART-S |
| Ligtenberg et al. (2011) | LIG |
| Kuipers Munneke et al. (2015) | KM |
| Simonsen et al. (2013) | SIM |
| CROCUS (Vionnet et al., 2012) | CRO |
| Morris and Wingham (2014) | MW |

instantaneous accumulation rate at a given time step. The mean accumulation rate for any layer of firn at time $t$ and depth $z$ with $Age_{z,t}$ is determined by the integrated accumulation-rate $\dot{b}$ history (Li and Zwally, 2011, 2015):

$$\bar{\dot{b}}(z,t) = \frac{1}{Age_{z,t}} \int_{t-Age_{z,t}}^{t} \dot{b}(t')dt'$$ (3)

The CFM uses $\bar{\dot{b}}$ because a firn-densification model dependent on the instantaneous accumulation rate will predict that no densification occurs when the accumulation rate is zero (Li and Zwally, 2011), which is not realistic. This approach may be different than how some of the models were originally formulated, and the CFM includes an option to use the instantaneous accumulation rate. In steady state, the mean accumulation rate is the same as the instantaneous rate.

The surface density $\rho_s$ of a new layer of firn in the CFM can be a constant value or can vary in time. In the case of time-varying $\rho_s$, it is determined by a parameterization (e.g. Kuipers Munneke et al., 2015) or by randomly selecting a value from a specified distribution.

We have coded each of the firn-densification models in the CFM as we have interpreted their descriptions in their original publications, and we have corrected any known errors. We next provide a basic description of each of the models and any nuances associated with coding them. The subsection headings also include the abbreviations that we use for each model in the application described in Section 3. For additional descriptions of the firn-densification models included in the CFM, see the original publications.





### 2.2.1 Herron and Langway (1980, *HL*)

Herron and Langway (1980) is a benchmark firn-densification model (Lundin et al., 2017); nearly all firn-densification models developed since 1980 are based in part on assumptions made by those authors. They used Sorge's Law and depth-density data from 17 firn cores to derive a widely-applicable firn-densification-rate equation. The CFM includes three formulations of the Herron and Langway (1980) model, which are detailed in Lundin et al. (2017): a "dynamic" model, a "stress-based model", and an "analytic" model. In steady state, all three give the same result, but the outputs vary in transient simulations. The CFM

uses the analytic model to generate an initial condition. Here we describe only the "dynamic" model, which is used in the application in Section 3.

Two assumptions in the Herron and Langway (1980) model have been used in numerous other firn-densification models. They are: (1) the change in porosity is linearly related to the stress change resulting from new snow accumulation (i.e. the densification rate is a function of the porosity; Schytt, 1958; Robin, 1958); and (2) the firn's densification rate has an Arrhenius

dependence on temperature. These assumptions can be incorporated into a densification-rate equation:

$$\frac{d\rho}{dt} = c(\rho_{ice} - \rho),\tag{4}$$

with

$$c = k \exp\left(-\frac{Q}{RT}\right)\dot{b}^a,\tag{5}$$

where $k$ and $a$ are constants, $Q$ is the Arrhenius activation energy (kJ mol$^{-1}$), $R$ is the gas constant (8.314 kJ mol$^{-1}$ K$^{-1}$), and

$T$ is the temperature (K). For *HL*, $c$ in Eq. (4) is given by:

$$
\begin{aligned}
c &= c_0 = 11 \exp\left(-\frac{10.16}{RT}\right)\dot{b}^{1.0} && (\rho \leq 550\,\mathrm{kg\,m^{-3}}) \\
  &= c_1 = 575 \exp\left(-\frac{21.4}{RT}\right)\dot{b}^{0.5} && (\rho > 550\,\mathrm{kg\,m^{-3}}).
\end{aligned}
\tag{6}
$$

*HL* uses units m water eq. a$^{-1}$ for $\dot{b}$. $T$ in the original model was the mean annual site temperature $T_m$, but since its establishment *HL* has also been implemented such that $T$ is the temperature of a specific parcel of firn. The parameters $k$, $a$, and $Q$ were all tuned to best fit the firn-core data. The activation energy derived by Herron and Langway (1980) is lower than most

other models, which causes it to be less sensitive to sub-annual temperature variability. We note that because of the different values of the exponent $a$ on $\dot{b}$, $k$ in Eq. (5) has different units for zones 1 and 2. The units for $\rho$ in *HL* are Mg m$^{-3}$, which are numerically equivalent to units g cm$^{-3}$.





#### 2.2.2 Barnola et al. (1991, *BAR*)

The Barnola et al. (1991) model was developed for ice-core $\Delta$age calculations. It uses the Herron and Langway (1980) model
for zone-1 densification. For $\rho > 550\,\mathrm{kg\,m^{-3}}$, the densification rate is given by

$$\frac{d\rho}{dt} = \rho_i A_0 \exp\left(\frac{-Q}{RT}\right) f \sigma_{eff}^n, \tag{7}$$

where $A_0 = 2.54 \times 10^4\,\mathrm{MPa^{-3}\,s^{-1}}$, the activation energy $Q$ is $60\,\mathrm{kJ\,mol^{-1}}$, $\sigma_{eff}$ is the effective stress (in MPa), and $n = 3$.
For zone-2 densification, Barnola et al. (1991) derived an equation empirically to match the densification rate and its derivative
at the zone-1/zone-2 and bubble close off transitions, and $f$ is given by

$$f = 10^{\alpha\rho^3 + \beta\rho^2 + \delta\rho + \gamma} \qquad (550 \leq \rho \leq 800\,\mathrm{kg\,m^{-3}}), \tag{8}$$

with $\alpha = -37.455$, $\beta = 99.743$, $\delta = -95.027$, and $\gamma = 30.673$. Beyond the $800\,\mathrm{kg\,m^{-3}}$ density horizon, $f$ is taken from
Pimienta (1987):

$$f = \frac{3}{16}(1 - \rho/\rho_i)/(1 - (1 - \rho/\rho_i)^{1/3})^3 \qquad (\rho > 800\,\mathrm{kg\,m^{-3}}). \tag{9}$$

#### 2.2.3 Arnaud et al. (2000) and Goujon et al. (2003, *GOU*)

Arnaud et al. (2000) developed a densification model based upon descriptions of grain-scale physical processes in firn, and
Goujon et al. (2003) extended that model by adding a heat-diffusion component. These models were developed for ice-core
delta-age reconstructions in Antarctica. In the CFM we refer to this family of models as the Goujon model, because the CFM
includes a heat-diffusion module. For zone-1 densification, the Goujon et al. (2003) model is based on grain-boundary sliding
work by Alley (1987). It describes the densification rate in zone 1 as:

$$\frac{dD}{dt} = \gamma\left(\frac{P}{D^2}\right)\left(1 - \frac{5}{3}D\right) \qquad (\rho \leq 550\,\mathrm{kg\,m^{-3}}) \tag{10}$$

where $D = \rho/\rho_{ice}$ is the relative density, $P$ is the overburden pressure (bar), and $\gamma$ is a scaling factor that depends on the
viscosity of grain boundaries and the geometry of the grains. It is notable that *GOU*'s densification rate in zone 1, unlike most
other firn-densification models, does not depend on temperature.

Goujon et al. (2003) base their description of zone-2 densification on sintering theory from Arzt (1982):

$$\frac{dD}{dt} = 4.1817 \times 10^4 \exp\left(-\frac{E_A}{RT}\right)(D^2 D_0)^{1/3}\left(\frac{a}{\pi}\right)^{1/2}\left(\frac{4\pi P}{3aZD}\right)^3 \qquad (\rho > 550\,\mathrm{kg\,m^{-3}}) \tag{11}$$

where $E_A$ is the activation energy, given as $60\,\mathrm{kJ\,mol^{-1}}$, $R$ is the gas constant, $T$ is the temperature (K), $a$ is the average
contact area between the grains relative to the initial grain radius, and $Z$ is the coordination number, i.e. the average number
of neighboring grains to a central grain in the firn crystalline structure. $D_0$ is the zone 1-zone 2 transition relative density.
Details on determining $a$ and $Z$ can be found in the original publications. Unlike other firn-densification models, which specify
a constant transition density, the Goujon and others (2003) model uses a transition density that depends on $T_m$ (K), given by

$$D_0 = 0.00226T_m + 0.03 \tag{12}$$





Goujon et al. (2003) specify that $\gamma$ in Eq. (10) should be set so that the densification rate is continuous at $D_0$.

Buizert et al. (2015) described an issue in implementing the Goujon et al. (2003) model, which we review here. In the event that $D_0 \geq 0.6$, $dD/dt$ given by Eq. (10) becomes zero for $D = 0.6$ and negative for $D \geq 0.6$, which is not realistic.

Additionally, at $D = D_0$, the densification rate predicted by Eq. (11) is infinite because the contact area $a$ equals zero. We avoid these issues in our implementation of *GOU* by doing the following. We limit the value of $D_0$ given by Eq. (12) to a maximum value of $0.59$, which occurs for temperatures greater than $\sim -25\,^\circ\mathrm{C}$. This value of $D_0$ corresponds to a density of $541\,\mathrm{kg\,m^{-3}}$, which results in *GOU* always predicting the zone 1 - zone 2 transition occuring at lower densities than the commonly-used value of $550\,\mathrm{kg\,m^{-3}}$. We follow the suggestion in Buizert et al. (2015) and put the zone 1 - zone 2 transition

at relative density $D_0' = D_0 + \epsilon$, where $\epsilon$ is a small number. The densification rate in zone 2 is still calculated using Eq. (11) using $D_0$ given by Eq. (12). We then iterate to find $\gamma$ in Eq. (10) that gives the maximum $dD/dt$ at the bottom of zone 1, with the condition that it does not exceed $dD/dt$ given by Eq. (11) at the top of zone 2.

### 2.2.4 Li and Zwally (2011, *LZ11*) & Li and Zwally (2015, *LZ15*)

The Li and Zwally (2011) and Li and Zwally (2015) are the latest in a lineage of models developed by the authors (Li and

Zwally, 2002, 2004). The models were developed to predict the surface-elevation changes associated with seasonal variability in accumulation and firn compaction rates. *LZ11* and *LZ15* are tuned to model firn in Greenland and Antarctica, respectively. Both share the same basic form:

$$\frac{d\rho}{dt} = \beta 8.36 \, (273.2 - T_K)^{-2.061} \bar{\dot{b}}(\rho_i - \rho), \tag{13}$$

where $T_K$ is the firn temperature as a function of time and depth with units Kelvin, and $\bar{\dot{b}}$ is the mean accumulation rate over

the lifetime of a parcel of firn (Eq. (3)) in $\mathrm{m\,water\,eq.\,a^{-1}}$. The difference between *LZ11* and *LZ15* is in the parameter $\beta$. For *LZ11*,

$$
\begin{aligned}
\beta = \beta_1 &= -9.788 + 8.996\dot{b}_m - 0.6165 T_{m,c} &\quad (\rho \leq 550\,\mathrm{kg\,m^{-3}}) \\
&= \beta_2 = \beta_1/(-2.0178 + 8.4043\dot{b}_m - 0.0932 T_{m,c}) &\quad (\rho > 550\,\mathrm{kg\,m^{-3}}).
\end{aligned}
\tag{14}
$$

For *LZ15*,

$$
\begin{aligned}
\beta = \beta_1 &= -1.218 - 0.403 T_{m,c} &\quad (\rho \leq 550\,\mathrm{kg\,m^{-3}}) \\
&= \beta_2 = \beta_1(0.792 - 1.080\dot{b}_m + 0.00465 T_{m,c}) &\quad (\rho > 550\,\mathrm{kg\,m^{-3}}).
\end{aligned}
\tag{15}
$$

where $T_{m,c}$ is the mean annual surface temperature in units Celsius and $\dot{b}_m$ is the long-term accumulation rate at the site being modeled.

*LZ11* and *LZ15* predict unrealistically high densification rates for firn near the freezing temperature (and is infinite at the freezing temperature), which makes them unsuitable for simulations of wet firn.





### 2.2.5 Helsen et al. (2008, *HEL*)

The Helsen et al. (2008) model was developed to simulate firn-column thickness changes in Antarctica to improve ice-sheet mass-change estimates derived from satellite-altimetry observations. Its development was based on the work of Li and Zwally (2002) and uses the same general form for its densification equation (Eq. (13)). Helsen et al. (2008) used additional firn core data from Antarctica to derive a different value for $\beta$. *HEL* uses a single $\beta$ for zone 1 and zone 2 densification, given by:

$$\beta = \beta_1 = \beta_2 = 76.138 - 0.28965 T_m. \tag{16}$$

The mean annual surface temperature $T_m$ in Eq. (16) has units K.

### 2.2.6 Arthern et al. (2010, *ART-T & ART-S*)

Arthern et al. (2010) derived a firn-densification model using firn-compaction-rate data from several sites in the Filchner-Ronne sector of Antarctica. The model is notable because it was the first firn-densification model to be based on compaction-rate measurements rather than upon a derived compaction rate from Sorge's Law. The authors also identified two processes in firn,

diffusion of water molecules through the ice lattice and grain growth, that have different activation energies. They hypothesized that these processes acting in concert result in a lower effective activation energy, which could explain the low temperature sensitivity in *HL*.

Arthern et al. (2010) describe two implementations of their model: the first is the complete transient, dynamical model described in their appendix, which we refer to as *ART-T*. It includes equations for densification rate based on evolving stress $\sigma$

and grain radius $r$ (based on the work of Gow et al. (2004)):

$$
\begin{aligned}
\frac{d\rho}{dt} &= k_c (\rho_i - \rho) \exp(-E_c/RT) \sigma / r^2 \\
\frac{dr^2}{dt} &= k_g \exp(-E_g/RT) \\
\frac{d\sigma}{dt} &= \dot{b} g
\end{aligned}
\tag{17}
$$

where $k_c$ and $k_g$ are empirically-derived constants. The water-molecule-diffusion activation energy $E_c$ is $60 \, \mathrm{kJ \, mol^{-1}}$, and the grain-growth activation energy $E_g$ is $42.4 \, \mathrm{kJ \, mol^{-1}}$. *ART-T* is not in common use; it is sensitive to the surface grain size, which is poorly constrained for model simulations over the range of climates encountered on ice-sheet scales.

Arthern et al. (2010) use several simplifying assumptions, including that of steady accumulation, to derive the second implementation of their model, which is the model presented in their main text. We refer to this implementation as *ART-S*. The





densification equations for zone-1 and zone-2 use the same form as the Herron and Langway densification equation (Eq. (4)), with parameters $c$ given by:

$$c_0 = 0.07\,\dot{b}g\exp\left(-\frac{E_c}{RT}+\frac{E_g}{RT_m}\right) \qquad (\rho \leq 550\,\mathrm{kg\,m^{-3}})$$

$$c_1 = 0.03\,\dot{b}g\exp\left(-\frac{E_c}{RT}+\frac{E_g}{RT_m}\right), \qquad (\rho > 550\,\mathrm{kg\,m^{-3}})$$

(18)

with $\dot{b}$ the mass accumulation rate (units $\mathrm{kg\,m^{-2}\,a^{-1}}$) and $T_m$ has units K. *ART-S* forms the basis of the models described by Ligtenberg et al. (2011), Simonsen et al. (2013), and Kuipers Munneke et al. (2015).

### 2.2.7   Ligtenberg et al. (2011, *LIG*) & Kuipers Munneke et al. (2015, *KM*)

The Ligtenberg et al. (2011) and Kuipers Munneke et al. (2015) models were developed to simulate firn densification in Antarctica and Greenland, respectively, using outputs from the regional climate model RACMO (Noël et al., 2018; van Wessem

et al., 2018). Their development was based on *ART-S*, but the authors used firn-core data to widen *ART-S*' applicability across the ice sheets. *LIG* and *KM* are the same as *ART-S* with the exception that $c_0$ and $c_1$ in Eq. (18) are multiplied by additional tuning coefficients. For *LIG*,

$$c_0^{(LIG)} = [1.435 - 0.151\ln(\dot{b})]\,c_0^{(ART-S)}$$

$$c_1^{(LIG)} = [2.366 - 0.293\ln(\dot{b})]\,c_1^{(ART-S)},$$

(19)

and for *KM*,

$$c_0^{(KM)} = [1.042 - 0.0916\ln(\dot{b})]\,c_0^{(ART-S)}$$

$$c_1^{(KM)} = [1.734 - 0.2039\ln(\dot{b})]\,c_1^{(ART-S)}.$$

(20)


For *LIG* and *KM*, $\dot{b}$ has units $\mathrm{kg\,m^{-2}\,a^{-1}}$. Ligtenberg et al. (2011) specified a densification rate as a function of both the firn temperature and mean-annual surface temperature, as is done in *ART-S* (Eq. (18)). Kuipers Munneke et al. (2015) used $T$ rather than $T_m$ in the grain-growth term of the Arrhenius factor; Steger et al. (2017) modified this to use $T_m$, and we include this latest version in the CFM.

### 2.2.8   Simonsen et al. (2013, *SIM*)

The Simonsen et al. (2013) model was developed for ice-sheet mass-balance studies. The authors used a Monte-Carlo inverse method with radar layers and regional-climate-model data to tune the parameters in the firn-densification model. Like *LIG* and *KM*, *SIM* also uses the densification equation from *ART-S* as a basis and multiplies $c_0$ and $c_1$ in Eq. (18) by additional factors:

$$c_0^{(SIM)} = f_0\,c_0^{(ART-S)}$$

$$c_1^{(SIM)} = f_1\frac{61.7}{\dot{b}^{0.5}}\exp\left(\frac{-3800}{RT_m}\right)c_1^{(ART-S)}.$$

(21)





Simonsen et al. (2013) found that the parameters $f_0$ and $f_1$ needed to be tuned and/or specified to simulate the firn at a particular site (see Lundin et al., 2017, Appendix); the CFM uses default values $f_0 = 0.8$ and $f_1 = 1.25$ (S. Simonsen, pers. comm., 2015).

### 2.2.9  *CROCUS* (Brun et al., 1992; Vionnet et al., 2012), *CRO*

The Crocus model was developed for mountain snowpacks (Brun et al., 1992), but it has also been used to simulate firn
densification and hydrology (e.g. Langen et al., 2017; Verjans et al., 2019). Its equations are also used for the subsurface scheme in the RCM MAR (Cullather et al., 2016), which is used to simulate surface mass balance of the Greenland and Antarctic ice sheets (Fettweis et al., 2017; Agosta et al., 2019; Alexander et al., 2019).

The Crocus model gives compaction in terms of a constitutive equation, relating stress $\sigma$ to the densification rate with a viscosity $\eta$ (Vionnet et al., 2012):

$$\frac{d\rho}{dt} = \frac{\rho\sigma}{\eta}$$

$$\eta = f_1 f_2 \eta_0 \frac{\rho}{c_\eta} \exp\left(a_\eta(T_{melt} - T) + b_\eta\rho\right) \tag{22}$$

where $\rho$ is the density, $T_{melt}$ is the melting temperature, $\eta_0 = 7.62237 \times 10^6\,\mathrm{kg\,s^{-1}\,m^{-1}}$, $a_\eta = 0.1\,\mathrm{K^{-1}}$, $b_\eta = 0.023\,\mathrm{m^3\,kg^{-1}}$. $f_1$ and $f_2$ are snow-viscosity correction factors. $f_1$ accounts for viscosity differences due to presence of liquid water; it is set to 1 in the CFM for model simulations in the dry-firn zone. $f_2$ accounts for angular grains. Following Langen et al. (2017) and van Kampenhout et al. (2017), $f_2$ is set to 4 in the CFM. Vionnet and others (2012) give $c_\eta = 250\,\mathrm{kg\,m^{-3}}$, but also following
van Kampenhout et al. (2017), we set $c_\eta = 358\,\mathrm{kg\,m^{-3}}$ in the CFM.

### 2.2.10  Morris and Wingham (2014, *MW*)

Morris and Wingham (2014) used a neutron probe to measure firn density in boreholes in successive years at several sites on the Greenland ice sheet. The high vertical spatial resolution of these measurements allowed the authors to measure both the strain between firn layers and density changes in those layers. They used this information to derive a compaction equation for
dry firn of density less than $550\,\mathrm{kg\,m^{-3}}$:

$$\dot{\epsilon} = \frac{k_0^*}{\rho_w g}\left(\frac{\rho_i - \rho}{\rho}\right)\left(1 - \bar{M}_0 m\right)\frac{1}{H(\tau)}\exp\left(-\frac{E_\alpha}{RT}\right)\sigma, \tag{23}$$

where $H(\tau)$ is a "temperature-history function":

$$H(\tau) = \int_{\tau_0}^{\tau} \exp\left(-\frac{E_\alpha}{RT(\tau')}\right)d\tau'. \tag{24}$$

In the above equations, $\rho_i$ and $\rho_w$ are the densities of ice and water, respectively, $k_0^*$ is a densification constant, and $\tau$ is the
age of a parcel of firn. $\tau_0$ is the age of the firn after it leaves the thin surface snow layer, which we take in the CFM to be zero. $m$ is the normalized deviation of $\rho(z)$ from a quadratic curve fit to the density profile, and $\bar{M}_0$ is a scaling constant. The CFM





**Table 2.** $(E_\alpha - E_H)$ in $\mathrm{kJ\,mol^{-1}}$, revised from Table 2 of Morris and Wingham (2014).

| Site | $T_m^*$ | $\bar{\bar{b}}$ | $E_\alpha = 60$ | $E_\alpha = 110$ | $E_\alpha = 200$ |
| --- | --- | --- | --- | --- | --- |
| | °C | $\mathrm{m\,w.e.\,a^{-1}}$ | $\mathrm{kJ\,mol^{-1}}$ | $\mathrm{kJ\,mol^{-1}}$ | $\mathrm{kJ\,mol^{-1}}$ |
| South Pole | -51.0 | 0.07 | 2.7 | 6.5 | 13.9 |
| T41D | -30.8 | 0.22 | 1.5 | 4.0 | 9.7 |
| Roi Baudouin | -15.0 | 0.38 | 0.8 | 2.6 | 7.1 |

by default uses the preferred activation energy $E_\alpha$ presented in Morris and Wingham (2014) of $110\,\mathrm{kg\,m^{-3}}$, though the authors also test their model with values of 60 and $200\,\mathrm{kJ\,mol^{-1}}$. As such, the CFM is coded to allow the user to vary $E_\alpha$ easily.

Morris and Wingham (2014) described simplifying assumptions for their model, which allowed them to use the same den-
sification constant $k_0^* = 11\,\mathrm{m\,w.e.^{-1}}$ as Herron and Langway (1980). However, this description was based on an error in the calculation of the history function from their data. Conceptually, the result of the error is that the authors' original hypothesis of a single densification process does not hold. In practice, this error causes the model to predict unrealistic densification rates. Table 2 shows the corrected values of $E_\alpha - E_H$ (revising Table 2 from Morris and Wingham (2014)), which can be used to calculate the corrected value of $k_0^*$ with the following equation:

$$k_0^* = 11\exp\left(-\frac{E^* - (E_\alpha - E_H)}{RT_m}\right),\qquad(25)$$

where $E^*$ is the Herron and Langway (1980) activation energy.

The error and its correction was described in a personal communication with E. Morris in April 2019; that communication is included with the CFM's documentation. *MW* only specifies densification rates for zone 1; the CFM is coded to use the Herron and Langway (1980) equation for zone-2 densification.

## 2.3 Temperature Evolution

The temperature in firn evolves by diffusion and advection. Heat diffusion in the CFM is modeled using a fully-implicit finite-volume scheme (Patankar, 1980). Advection of heat is inherently handled by the Lagrangian scheme. The CFM uses a Dirichlet (prescribed temperature) boundary condition at the surface; the surface temperature is set by the input forcing data. At the bottom of the domain the CFM uses a Neumann (i.e. prescribed gradient, set to zero) boundary condition for the
temperature by default; this condition can be easily adapted to use a non-zero gradient or a Dirichlet boundary condition.

Simulating heat diffusion in firn requires knowledge of the thermal conductivity, but there is no universally agreed upon parameterization for the thermal conductivity of snow and firn. The CFM includes a number of parameterizations for the thermal conductivity that have been published previously. Those are: Anderson (1976), Brandt and Warren (1997), Jiawen et al. (1991), Lüthi and Funk (2001), Riche and Schneebeli (2013), Schwander et al. (1997), Schwerdtfeger (1963), Sturm
et al. (1997), Van Dusen (1929), and Yen (1981).





In the wet-firn zone of an ice sheet, there is additional heat transport due to advection of liquid water at $0°C$ through the firn's pore space and due to latent heat release from meltwater refreezing. The CFM simulates the advective component with a meltwater percolation scheme, which is described in Section 2.6. The latent heat from refreezing is handled in one of two ways, depending on the meltwater percolation scheme that is used. The first uses a fully-implicit, finite-volume, enthalpy-diffusion
scheme to resolve latent-heat release and heat diffusion (Voller et al., 1990). We note that a similar enthalpy-based method was employed by Meyer and Hewitt (2017). The second computes latent heat release in the meltwater percolation scheme and separately uses the dry-firn heat-diffusion scheme; details of it are provided in Verjans et al. (2019).

### 2.4 Water-Isotope Diffusion

The CFM includes a module that calculates the diffusion of water isotopes, which occurs due to sublimation and deposition of
water molecules in the firn. This process is important to the interpretation of ice-core records. Variability in the water-isotopic composition of snow crystals that fall on the surface is damped by a diffusion process as the snow advects downward through the firn column. In the vapor phase, water molecules diffuse through the firn column's interconnected pore space, smoothing the highest-frequency variations in the vertical water-isotope profile. This diffusion process stops at the BCO depth where water vapor can no longer move through pore space. The total amount of diffusion that occurs in the firn column depends
on both the time it takes a parcel of firn to advect from the surface to the BCO depth and temperature of the firn during that time. In analysis of water-isotope records from ice cores, understanding this process of diffusion allows both for correction of high-resolution water isotope records and interpretation of past firn conditions recorded by the ice core (Gkinis et al., 2014; Jones et al., 2017).

The CFM isotope-diffusion module uses the equations presented in Johnsen et al. (2000). Each layer is assigned a water
isotope value at the surface. At each time step, the water isotopes diffuse according to Fick's second law. Both $\delta^{18}O$ and $\delta D$ can be tracked, accounting for differences in fractionation factors and air diffusivities. This module can be used to study cumulative water-isotope diffusion under a range of firn conditions and/or to simulate water-isotope records to compare to deep ice core records. The CFM also tracks diffusion length.

### 2.5 Firn-air Diffusion

The CFM includes a firn-air diffusion module coupled to the firn-densification model. Previous modeling work has considered firn densification and firn-air transport separately; that is, firn-air models have assumed steady-state firn depth-density and effective-diffusivity profiles. The firn-air module allows us to simulate gas transport while simultaneously modeling evolution of firn depth and density in a changing climate.

The CFM firn-air module solves the firn-air equation (Severinghaus et al., 2010; Birner et al., 2018):

$$\frac{DC}{Dt} = \frac{1}{\phi_{op}} \frac{D}{Dz} \left[ \phi_{op,z}(t) D_{eff}(t,z) \left( \frac{dC}{dz} - \frac{\Delta m \, g}{RT} + \Omega \frac{dT}{dz} \right) \right] + w_{air}(z) \frac{\partial C}{\partial z} \quad (26)$$

where $C$ is the concentration (ppm or ppt) or delta value (‰) of a gas species. $\phi_{op}(t,z)$ is the open porosity (unitless). $\Delta m$ is the molar mass difference between two isotopologues or the molar mass difference from air ($\mathrm{kg\,mol^{-1}}$). $\Omega$ is the thermal-





diffusion sensitivity ($\mathrm{K}^{-1}$), which is specific to individual gases (Severinghaus et al., 2001). $w_{air}(z)$ is the advection rate of the air relative to the pore matrix. $D_{eff}(t,z)$ is the effective diffusivity; as porosity of firn decreases, molecules must take a

longer and more tortuous path in order to diffuse; the effective diffusivity accounts for this by scaling the free-air diffusivity, $D_{FA}$, with the tortuosity, $\tau$, of the firn (Buizert et al., 2012): $D_{eff} = D_{FA}/\tau$. In this work we take the term diffusivity to mean effective diffusivity.

    Equation (26) is solved on the CFM's Lagrangian grid, which causes the downward advection of gasses to be handled by the downward-moving reference frame. However, air advects downward slower than the surrounding firn (upward relative to the

downward-moving grid) because the densification of the firn increases the air pressure in the open porosity (i.e. the air-pressure gradient is positive downward, causing $w_{air}(z)$ to have a negative sign in a Lagranian framework). Previous Lagrangian firn-air models have ignored this effect (e.g. Trudinger et al., 1997); the CFM has the option to ignore it or to include it through the description provided by Buizert (2011).

    The CFM also allows the user to choose which parameterization for diffusivity to use. As of the time of writing, the CFM

includes the diffusivity parameterizations published by Schwander et al. (1988), Battle et al. (1996), Severinghaus et al. (2001), Freitag et al. (2002), Witrant et al. (2012), and Adolph and Albert (2014).

    Modeling the diffusivity and the close-off physics requires knowing the BCO depth. To find the corresponding close-off density $\bar{\rho}_{co}$, the CFM uses the relationship published by Martinerie et al. (1994):

$$\bar{\rho}_{co} = \left( \frac{1}{\rho_{ice}} + 6.95 \times 10^{-7} T_m - 4.3 \times 10^{-5} \right)^{-1} \tag{27}$$

with temperature in K and $\rho$ in $\mathrm{kg\,m^{-3}}$. Bubbles close off over a range of densities (and therefore depths). For example, Schwander and Stauffer (1984) reported that at Siple Station, 80% of bubbles close off between 795 $\mathrm{kg\,m^{-3}}$ and 830 $\mathrm{kg\,m^{-3}}$. Equation (27) predicts the mean close-off density (and thus mean close-off porosity $\phi_{co}$). The density of full bubble close off, $\rho_{CO}$, where $\phi_{op} = 0$, will be slightly greater than $\bar{\rho}_{co}$, and its depth $z_{co}$ slightly deeper than $z(\bar{\rho}_{co})$.

    Equation (26) also requires knowing the open porosity $\phi_{op}$, which equals the total porosity $\phi_{tot}$ minus the closed porosity

$\phi_{cl}$. The CFM uses the parameterization for $\phi_{cl}$ as a function of $\phi_{tot}$ presented in Goujon et al. (2003):

$$\phi_{cl} = 0.37\phi_{tot} \left( \frac{\phi_{tot}}{\phi_{co}} \right)^{-7.6}. \tag{28}$$

    Finally, the CFM's firn-air module has an option to specify the lock-in depth (LID), the depth at which gravitational enrichment ceases despite the existence of open porosity. The lock-in zone (LIZ) is the zone between the LID and the close-off depth. The LIZ is not well understood, but recent work has suggested that it may be created by the firn's three-dimensional layer

structure and by barometric pumping (Birner et al., 2018). In the CFM, the diffusivity below the specified LID is set to zero to inhibit further gas diffusion, and the lock-in density is determined by subtracting $14\,\mathrm{kg\,m^{-3}}$ from $\bar{\rho}_{co}$ (Blunier and Schwander, 2000).

    The CFM does not include certain features that some firn-air models include (Buizert et al., 2012) such as bubble trapping rate, bubble pressure, total air content, dispersive mixing in the LIZ, and the mean and distribution of gas ages in the closed

porosity. These features will be integrated into future releases of the CFM.




## 2.6 Melt

The CFM has several meltwater-percolation schemes to choose from, including two "tipping-bucket schemes", a Richards
Equation single-domain scheme and a Richards Equation dual-domain approach. The latter two and one of the bucket schemes
are described in Verjans et al. (2019). The second bucket scheme is similar to other bucket schemes that have been developed:
at each time step, the volume of surface meltwater is allowed to percolate downward through the pore space. As the water
reaches each model node (i.e. parcel of firn) in the model grid, the CFM first calculates the volume of water that refreezes
due to the firn's cold content (the energy required to bring the firn's temperature to $T_{melt}$), and that volume is immediately
refrozen. The temperature of that parcel becomes the freezing temperature. Then, the volume of liquid that stays in the parcel
due to capillary action (Coléou and Lesaffre, 1998) is subtracted from the meltwater volume. The remaining liquid moves
downward to the next volume; this process continues until the entire volume of meltwater is accounted for. In the event that
the firn's cold content can freeze the entire volume of meltwater, the firn temperature is raised by an amount that equals the
latent heat released by refreezing. If the meltwater encounters an impermeable layer, which we define as a layer with a density
$\geq 800\,\mathrm{kg\,m^{-3}}$ (Gregory et al., 2014), the water fills in the pore space in the parcel(s) above and remains liquid. After this
percolation routine, the CFM solves for temperature in the entire firn column using the enthalpy scheme described in Section
2.3 and calculates the new mass and density of each parcel.

## 2.7 Grain Growth

The CFM can optionally simulate grain size (assuming spherical grains) during a model run using one of two parameterizations.
The first gives the change in mean grain radius $r$ (m) as Gow (1969):

$$\frac{dr}{dt} = \frac{1}{2r} k_g \exp(-E_g/RT) \tag{29}$$

with grain-growth activation energy $E_g = 42.4\,\mathrm{kJ\,mol^{-1}}$ and constant $k_g = 1.3\times10^{-7}\,\mathrm{m^2\,s^{-1}}$ taken from Cuffey and Paterson
(2010, p. 40) and Arthern et al. (2010), respectively. Eq. (29) is the grain-growth equation used for *ART-T*.

The second grain growth parameterization accounts for the effect of liquid water on grain metamorphism (Section 2.6;
Verjans et al., 2019). It is taken from Katsushima et al. (2009), who used equations from Tusima (1978) and Brun (1989) to
simulate water flow in seasonal snowpacks. In the CFM, it is implemented as:

$$\frac{dr}{dt} = \frac{1}{8\times10^9 r^2} \times \min\left[\frac{2}{\pi}\left(1.28\times10^{-8}+4.22\times10^{-10}\theta^3\right), 6.94\times10^{-8}\right], \tag{30}$$

where $\theta$ is mass-percent liquid water content.

As previously mentioned, the surface grain size of polar firn across the range of climates on ice sheets is not well constrained.
For Eq. (29), the CFM uses a uniform surface grain size $r_0 = 0.1\mathrm{mm}$. For Eq. (30), the CFM uses the empirical formula for
surface grain size as a function of $T_m(°C)$ and $\dot{b}\,(\mathrm{m\,water\,eq.\,a^{-1}})$ given by Linow et al. (2012):

$$r_0 = b_0 + b_1 T_m + b_2 \dot{b} \tag{31}$$

with constants $b_0 = 0.781$, $b_1 = 0.0085$, and $b_2 = -2.79$.





The CFM's grain-growth module can be easily adapted to include other grain-growth formulations. Equations describing evolution of other grain-scale physical properties, such as the specific surface area, could also be easily integrated into the CFM's framework if future research provides insights into those properties.

**CFM APPLICATIONS**

We demonstrate the utility of the CFM in two model applications. In the first, we compare the outputs of 13 firn-densification models when forced with accumulation rates and temperatures predicted by a Regional Climate Model (RCM). In the second, we use the coupled firn-density firn-air model to simulate concentrations of gas stable isotopes trapped in ice cores during rapid climate changes.

**3    Model Application 1: Intercomparison of firn-model outputs at Summit, Greenland**

In our first model application, we investigate uncertainty in firn-model outputs that results from the choice of firn-densification model. This work follows the Firn Model Intercomparison Experiment (FirnMICE; Lundin et al., 2017), which compared the responses of eight firn-densification models to synthetic climate histories that featured step changes in temperature and accumulation rate. Here, we use the CFM to expand upon that work by comparing outputs from 13 different firn-densification

models forced with temperature and accumulation histories for Summit, Greenland ($72.58°$ N, $38.48°$ W, $3200$m). The mean annual temperature at Summit is $-31.4°$C, and the annual accumulation averages $0.23\,\mathrm{m\,ice\,eq.\,a^{-1}}$. Historically, Summit has rarely experienced melt. Summit was the site of the GISP2 ice cores.

The FirnMICE project featured results submitted by different research groups running their own firn-densification model codes; here, we run the firn densification models within the CFM framework. This allows us to compare the outputs from

different firn-densification models without concern of artifacts associated with different numerical methods (e.g. grid size, temperature-diffusion scheme, etc.); that is, differences in model outputs are due to differences in the particular representation of physics in each firn-densification model.

Sources of uncertainty in firn-model outputs include the surface boundary conditions (i.e. the forcing) and the representation of physical processes in the firn-densification model (i.e. the algorithms). We can begin to understand these uncertainties by

comparing different outputs produced by a single firn-densification model forced with range of plausible inputs, or by comparing outputs from different firn-densification models when they are forced by the same initial and boundary conditions. In this application we use the latter approach to leverage the CFM's ability to run multiple firn-densification models. In particular, we examine the variability in model outputs that arises from firn-densification model choice using the metrics of depth-integrated porosity (DIP), surface-elevation change (dH), and bubble-close-off (BCO) age and depth. In order to avoid picking a "best"

firn-densification model, which would only be "best" at our single test site, we focus on comparing the models' outputs to each other. We include data from a firn core drilled at Summit in 2007 as a reference point.





### 3.1 Methods

We forced each of the firn-densification models in the CFM (Table 1) with skin temperature and surface mass balance (SMB) outputs for Summit, Greenland from the RCM MAR3.9 (Fettweis et al., 2017). We used the MAR products derived from ERA-40/ERA-Interim, which begin in January 1958 and end in October 2018. We ran the CFM at monthly time steps. The domain extended from the surface to $\sim 220$ m depth; the exact depth varied by model because the CFM's Lagrangian framework uses a fixed number of model nodes rather than a fixed grid. The predicted densities at that depth also varied by model but were generally $916\,\mathrm{kg\,m^{-3}}$ to $917\,\mathrm{kg\,m^{-3}}$. The surface density was held constant at $300\,\mathrm{kg\,m^{-3}}$. For these simulations, we did not use the melt module and thus ignored any melt that may have occurred in the forcing fields during the simulation period.

In order to run a firn-densification model, the model must first be spun up to an appropriate initial condition. Here, the initial condition that we desire is a firn depth-density-temperature profile for the start of the year 1958. Ideally, the spin-up process would produce a firn-density and firn-temperature profile that was the actual value of the profile in 1958 (i.e. what would have been measured in the field). At Summit, and most sites, that is not possible, so the goal is to create an initial condition that is as representative as possible of the firn at that time. To do this, we generated temperature and accumulation-rate histories for the 1000 years prior to the start of the model run (years 958 to 1957); we spun up the model for 1000 years because that was long enough to refresh the entire firn column (i.e. to remove any artifacts of the model initialization) and to ensure that the firn had come to thermal equilibrium. Similar to Kuipers Munneke et al. (2015), we assumed that the 1958 to 1978 climate was in steady state and is representative of the climate for the previous 1000 years. We spun up the CFM by repeating the 1958 to 1978 MAR temperature and SMB fields. The firn during spin up does not reach true steady state because there is climatological variability in the spin-up forcing data, but the firn does reach a state where the variability in its properties (e.g. porosity) is consistent with a steady-state climate that includes natural variability.

### 3.2 Model intercomparison metrics

We compare the model results using several metrics. The depth-integrated porosity, $DIP(z)$, is the volume of air contained within a meter-by-meter square firn column (units $\mathrm{m^3/m^2}$) above depth $z$, given by:

$$DIP(z) = \int\limits_0^z \phi(z')dz' = \int\limits_0^z \frac{\rho_i - \rho(z')}{\rho_i}dz' \tag{32}$$

where $\phi$ is the porosity, $\rho(z)$ is the density at $z$, and $\rho_i$ is the density of ice, taken in this work to be $917\,\mathrm{kg\,m^{-3}}$. DIP change is a key parameter used to convert volume-change measurements (e.g. surface-elevation change from altimetry) into mass change for sea-level-rise estimates (Ligtenberg et al., 2014). DIP is also called firn-air content (FAC).

When reporting the DIP predicted by a model, it is important to also report the maximum depth (and the corresponding density) to which the firn was modeled, because if the bottom of the model domain is shallower than the transition to ice, there will be additional porosity beyond the model domain. In this study, we compare the models' predicted DIP in the upper





15 m and 80 m and the total DIP (i.e. in the entire modeled firn column), which we refer to as $DIP_{15}$, $DIP_{80}$, and $DIP_{tot}$, respectively.

When comparing the outputs from the firn-densification models, one can consider both the inter-model differences in the predicted DIP and how the DIP changes through time in each model. The change in DIP is the quantity of interest for mass-balance studies that need to adjust surface-elevation measurements for interannual variability in firn thickness. Uncertainty in the total DIP is less consequential for these studies, but the total DIP is of interest when calculating the absolute mass of the ice sheet and predicting the volume of meltwater that can be retained on the ice sheet (e.g. Vandecrux et al., 2019).

Predicting surface-elevation change through time ($dH/dt$) due to firn processes is essential for making corrections to
surface-elevation measurements from altimetry to derive mass changes. We calculate $dH$ at each model time step by summing the ice-equivalent snow-accumulation rate $\dot{b}$, firn-compaction rate $v_{fc}$, vertical ice velocity $v_{ice}$ at the bottom of the firn column due to dynamic ice-sheet processes, and vertical bedrock-motion rate $v_{bed}$:

$$\frac{dH}{dt} = \dot{b} + v_{fc} + v_{ice} + v_{bed}. \tag{33}$$

In steady state, new-snow accumulation rate is equal to the combined firn-compaction and ice-sheet-thinning rates. In this
application, we assume that $v_{ice}$ is equal to the 1958-1978 mean ice-equivalent accumulation rate, which is in essence an assumption that the deep firn and ice sheet below are in steady state. We also assume for this application that $v_{bed}$ is zero and that additional layer thinning due to horizontal strain is negligible.

The BCO depth is the depth at which all porosity becomes closed (versus open and interconnected) and air is occluded in bubbles; the BCO age is the corresponding age of the firn at the BCO depth. The BCO density is commonly taken to be
$\sim 830\,\mathrm{kg\,m^{-3}}$, but in reality, the BCO depth does not correspond to an exact density. Nevertheless, here we use the $830\,\mathrm{kg\,m^{-3}}$ density horizon for model comparisons. The BCO age is of interest to the ice-core science community as BCO age is a key parameter for determining $\Delta$age.

For each of these metrics, we calculate the mean, the standard deviation $\sigma$, and coefficient of variation ($CV$; the ratio of $\sigma$ to the mean) of the models' results. There is no reason to believe that the mean of the models should give a better result (i.e.
closer to observations) than a particular model; however, these statistics are useful for understanding how the models compare to one another.

### 3.3 Model comparison results

#### 3.3.1 Depth-density and depth-DIP profiles

Figure 1 shows depth-density (left panels) and depth-DIP(z) (right panels) profiles predicted by the various models in October
2018, the end of the model runs. The upper and lower panels show the model results to 80 m and 15 m depth, respectively. Each of the panels in Fig. 1 are included as layered .pdf files in the supplementary files; the results from each model are plotted as a layer that can be toggled on and off using Adobe Reader or Acrobat software. Table 3 lists the models' predicted





**Figure 1.** Profiles of depth-density (left panels) and depth-DIP (right panels) predicted by the models listed in Table 3 in the upper 80 m (top panels) and 15 m (lower panels) of firn. Each panel is also included in the supplementary material as a layered .pdf file.

values of $DIP_{15}$, $DIP_{80}$, and $DIP_{tot}$; the depth and age of the $830\,\mathrm{kg\,m^{-3}}$ density horizon (columns **DEP830** and **AGE830**, respectively); and the linear trend of surface-elevation change for the last 10 years of the model run. Table 3 also shows the arithmetic mean ("model mean" row) and standard deviation ("model $\sigma$" row) of the model results. These rows have two values for several columns; the values in parentheses are the statistics excluding *CRO*, which predicts anomalously low density at greater depths.





Because the depth and density of the firn are different at the bottom of the domain for each firn-densification model, comparing the various models' $DIP_{tot}$ is not a direct comparison metric (as opposed to comparing $DIP_{15}$). However, the amount

of porosity that is in the bottom of the firn column is a very small percent of the total. For example, *BAR* reaches $916 \, \mathrm{kg \, m^{-3}}$ at 126 m and $917 \, \mathrm{kg \, m^{-3}}$ at 180 m; the DIP in this interval is 0.07 m, or 0.3% of $DIP_{tot}$. We thus consider $DIP_{tot}$ as a worthy metric for comparison.

Table 3 also includes DIP and BCO data derived from a firn core drilled at Summit in 2007 (Lomonaco et al., 2011). The firn-core density data have high variability with depth; we smoothed the depth-density data and report the values from that

smoothed depth-density profile. We estimate the age of the core using the depth-age scale from the GISP2 ice core (Meese, 1999).

Near the surface (zone-1 densification), the models show a variety of responses. The mean $DIP_{15}$ is 7.75 m with $\sigma = 0.81 \mathrm{m}$, which gives $CV = 10.5\%$. This mean is close (1.2%) to the value in the core (7.66 m). The DIP at 15 m depth ranges from 6.4 m (*LZ15*, *GOU*) to 8.9 m (*HEL*). Aside from those three models and *MW*, all of the models are within 10% of the data. If we

exclude *LZ15*, *GOU*, and *HEL*, which were all developed for Antarctica, the mean $DIP_{15}$ becomes 7.89 m with $\sigma = 0.57 \mathrm{m}$. However, it is important to note that many sites in West Antarctica have a climate similar to that at Summit. Additionally, the mean of the remaining 10 models is slightly farther from the value derived from the core. We expect the models to show similar results near the surface because they all start with the same surface boundary condition (i.e. $DIP(z = 0\mathrm{m}) = 0\mathrm{m}$ and $\rho_{surface} = 300 \, \mathrm{kg \, m^{-3}}$). Additionally, one may expect models to perform well near the surface in the dry firn zone because

there is a relative abundance of shallow firn cores from Greenland and Antarctica that can be used for model calibration (see e.g. Kuipers Munneke et al., 2015).

The depth-density results in zone 1 show how the models differ in their sensitivity to temperature. *MW* is the most sensitive to temperature, and it has the greatest density variability in zone 1. The *GOU* densification-rate equation for zone 1 is not a function of temperature, and *CRO* has relatively low temperature sensitivity. As such, these two models have very smooth

depth-density profiles.

The models diverge in their predictions through zone 2. *HEL* and *MW* predict the highest $DIP_{80}$, and *ART-S* predicts the lowest $DIP_{80}$. The mean of the models' $DIP_{80}$ is 20.8 m and $\sigma$ is 2.99 m, giving a $CV$ of 14.3%. The $DIP_{80}$ derived from the core is 22.7, a difference of 8% from the model mean.

Beyond 80 m, the models spread slightly more: excluding *CRO*, the $CV$ is 16%. The models should not be expected to spread

significantly beyond 80 m because there is relatively little porosity beyond that depth, and all of the models are formulated to prevent densification beyond the ice density; i.e. they effectively have a fixed density boundary condition at the bottom of the domain.

*CRO* is a notable outlier in the deep firn. It predicts densification that is significantly slower than the other models. Its $DIP_{80}$ is similar to the other models, but its density is $\sim 100 \, \mathrm{kg \, m^{-3}}$ less than the other models. *CRO*'s $DIP_{tot}$ is 40 m, nearly twice

the mean of the other models. This behavior is not surprising because *CRO* was developed as a seasonal snow model and therefore is not calibrated to accurately predict densification of higher-density firn. This is consistent with results from Lundin et al. (2017), who also found that a snow model did not predict densification well in deeper (zone 2) firn.





**Table 3.** Model results and firn core data, including DIP at 15 m and 80 m depth and the bottom of the model domain ($\sim 220 - 230$ m); the depth and age of the $830\,\mathrm{kg\,m^{-3}}$ density horizon; and the linear trend in surface elevation change (i.e. a regression of the model results shown in Fig. 2) in the last 10 years of the model run (2008 to 2018). The data in the CORE row are derived from a firn core drilled in 2007, and the depth-age scale of that core is estimated using the GISP2 timescale. The MODEL MEAN and MODEL $\sigma$ rows show the statistics for all 13 models; the value in parentheses are the statistics excluding *CRO*, which predicts anomalously low densification rates in the deeper firn.

| | $DIP_{15}$ | $DIP_{80}$ | $DIP_{tot}$ | $DEP_{830}$ | $AGE_{830}$ | $dH/dt_{08-16}$ | $dH/dt_{16-18}$ | $dH/dt_{08-18}$ |
| | (m) | (m) | (m) | (m) | (a) | (cm a$^{-1}$) | (cm a$^{-1}$) | (cm a$^{-1}$) |
|---|---|---|---|---|---|---|---|---|
| **CORE** | 7.66 | 22.7 | – | 79.52 | 238 | – | – | – |
| **HL** | 8.28 | 22.38 | 25.03 | 74.81 | 250 | -0.271 | 3.594 | 0.676 |
| **BAR** | 8.28 | 21.91 | 22.71 | 70.03 | 231 | -0.463 | 3.246 | 0.479 |
| **GOU** | 6.47 | 18.02 | 19.67 | 62.04 | 214 | -0.256 | 2.757 | 0.647 |
| **LZ11** | 7.95 | 21.48 | 24.01 | 72.75 | 246 | -0.148 | 3.300 | 0.792 |
| **LZ15** | 6.42 | 17.45 | 19.48 | 65.46 | 232 | 0.077 | 4.130 | 0.975 |
| **HEL** | 8.92 | 25.53 | 27.47 | 73.09 | 227 | 0.108 | 4.010 | 1.084 |
| **ART-S** | 6.99 | 15.57 | 16.14 | 49.47 | 167 | 0.280 | 2.675 | 1.061 |
| **ART-T** | 8.12 | 18.98 | 19.88 | 57.84 | 190 | -0.606 | 2.720 | 0.275 |
| **LIG** | 7.99 | 20.2 | 21.66 | 64.37 | 214 | 0.113 | 3.384 | 1.007 |
| **KM** | 8.24 | 23.09 | 26.17 | 78.32 | 262 | 0.352 | 3.342 | 1.191 |
| **SIM** | 7.47 | 18.24 | 19.38 | 59.42 | 201 | 0.147 | 2.916 | 0.952 |
| **CRO** | 6.96 | 23.05 | 40.6 | 174.74 | 662 | -0.043 | 3.187 | 0.691 |
| **MW** | 8.67 | 24.54 | 27.61 | 79.49 | 261 | 0.167 | 2.614 | 1.02 |
| **MODEL MEAN** | 7.75 | 20.8 | 23.83 | 75.53 | 258.37 | -0.0417 | 3.221 | 0.8347 |
| | | | (22.43) | (67.26) | (224.75) | | | |
| **MODEL $\sigma$** | 0.81 | 2.99 | 6.14 | 31.05 | 124.31 | 0.289 | 0.488 | 0.2676 |
| | | | (3.67) | (9.07) | (28.85) | | | |

The mean depth of the $830\,\mathrm{kg\,m^{-3}}$ density horizon (excluding *CRO*) is 67 m, compared to 80 m in the firn core, and the $\sigma = 9.07$m ($CV = 13\%$). The mean age is 224 years with $\sigma = 29$ years ($CV = 13\%$). A number of models do predict $DEP_{830}$

to within several meters, but these models all predict that $AGE_{830}$ is older than is observed. Likewise, the models that predict $AGE_{830}$ best predict $DEP_{830}$ that is too shallow. This consistent mismatch of BCO depths and ages may suggest that the sensitivities of the firn-densification models to accumulation rate and/or temperature are incorrect. For example, the models may have an underestimated temperature sensitivity but an overestimated $\dot{b}$ sensitivity. In this case, these faulty sensitivities compensate for each other; the models predict $d\rho/dt$ reasonably well and $AGE_{830}$ is about right. However, due to the overes-

timated $\dot{b}$ sensitivity, not enough material has been accumulated over the time required to reach a density of $830\,\mathrm{kg\,m^{-3}}$, and thus $DEP_{830}$ is too shallow.





### 3.3.2 Surface elevation change through time ($dH/dt$)

Figure 2 shows the modeled surface elevation through time predicted by the models for the last 10 years of the model runs (October 2008 to October 2018). It is also included as a layered .pdf file in the supplementary material. Table 3 lists the linear
least-squares trends ($\mathrm{cm\,a^{-1}}$) for each model from Octobers 2008-2016, 2016-2018, and 2008-2016 (columns $dH/dt_{08-16}$, $dH/dt_{16-18}$, and $dH/dt_{08-18}$, respectively). All the models predict that surface elevation has increased since 2008. The mean change from 2008 to 2018 is +12.5 cm, and the $\sigma = 2.7$cm ($CV = 21\%$).

In general, the models that predict the largest surface-elevation increase consistently predict the largest increase throughout the entire time series, and vice versa; i.e. the lines on Fig. 2 generally remain in the same location relative to one another.
However, there are times that certain models change their relative positions. For example, the surface-elevation increase since 2008 predicted by *ART-S* is the largest among the models as of mid-2016, but it is in the middle of the models at the end of the simulation. This behavior is a reflection of the models' different sensitivities to temperature, accumulation rate, and density.

Between 2008 and 2016, the change in surface elevation predicted by the models varies through time but does not deviate significantly from zero. The mean of the trends is $-0.042\,\mathrm{cm\,a^{-1}}$, and the models do not agree on the sign of the trend: six
predict a small negative trend and seven predict a small positive trend.

From October 2016 to the end of the model runs, the models all predict a surface-elevation increase; this occurs because there are numerous months in that time period when MAR predicts accumulation was higher than average. *LZ15* predicts the largest trend ($4.13\,\mathrm{cm\,a^{-1}}$), and *MW* the smallest ($2.61\,\mathrm{cm\,a^{-1}}$). The mean of the models' trends for these two years is $3.22\,\mathrm{cm\,a^{-1}}$ with $\sigma = 0.48\,\mathrm{cm\,a^{-1}}$ ($CV = 15\%$). The four models with the largest trend over this period (*LZ15*, *HEL*, *HL*, and *LIG*) were
all tuned using Antarctic firn cores. There is a trade off between sensitivity to temperature and sensitivity to accumulation rate involved in tuning a firn-densification model, and the larger trends predicted by these "Antarctic" firn models may indicate that models tuned specifically for Antarctica are biased towards sensitivity to accumulation rate. For example, between *LIG* and *KM*, which are twin models tuned for Antarctica and Greenland, respectively, *LIG* predicts larger densification rates for the same mean accumulation rate (Eqs. 19 and 20) at sites with accumulation less than $0.8\,\mathrm{m\,ice\,eq.\,a^{-1}}$. This bias could occur
if a large portion of the Antarctic cores came from sites with similar temperatures. Alternatively, models that are tuned for Greenland could be biased towards temperature sensitivity; e.g. *MW*, with the smallest trend over these two years, includes a significantly higher activation energy in the Arrhenius term.

Although the elevation-change trend is clearly not linear over the October 2008 to 2018 period, fitting a linear trend to the modeled elevation changes further illustrates the differences between the models. In this case, the mean trend is $0.83\,\mathrm{cm\,a^{-1}}$,
and $\sigma = 0.27\,\mathrm{cm\,a^{-1}}$ ($CV = 32\%$). *ART-T* predicts the smallest trend in $dH/dt$ ($0.28\,\mathrm{cm\,a^{-1}}$; the $CV$ drops to 24% if *ART-T* is excluded). *MW* predicts the smallest 2016-2018 trend, but it predicts the fourth largest for 2008-2018. *KM* predicts the largest 2008-2018 trend ($1.19\,\mathrm{cm\,a^{-1}}$), whereas it had only the fifth-largest trend for 2016-2018.

Collectively, these results indicate that though the magnitude of surface elevation changes is relatively small, the models do not agree well with one another when simulating firn evolution in response to climate variability. The models predict different





surface-elevation trends relative to one another depending on the period considered, and in periods of relatively small changes in surface elevation, fitting trends to the modeled elevation change yields different signs depending on the model chosen.

### 3.3.3  Firn Model Uncertainty

Our results show that the firn-densification model choice can be a significant source of uncertainty in applications requiring a firn model. Our goal with this application was to demonstrate the utility of the CFM in a simple model comparison exercise; as

such, we have avoided detailed comparisons to data and instead focus on the broad agreement among the models. We ran 13 models, and they do not agree within 10% when considering the DIP, BCO age and depth, or trend in surface-elevation change. Models that agree well using one metric do not necessarily agree with a different metric. For example, *KM* and *CRO* predict nearly the same $DIP_{15}$, but *KM*'s 10-year trend in $dH/dt$ is the highest of all models, and *CRO*'s is near the low end.

This is a challenge that the firn-modeling community continues to face: despite the number of firn-densification models

that have been proposed, no single model is widely accepted. The general form of the firn-densification models is relatively similar (e.g. $d\rho/dt$ is a function of $[\rho_i - \rho]$, which is an obvious "shut-off" to prevent over-densification), but they differ in their particular details (e.g. what the activation energy in the Arrhenius term is). Lundin et al. (2017) showed that firn-densification models do not agree when predicting steady-state or transient behavior when forced with synthetic climate, and our results corroborate those results.

Future work could include an analysis of uncertainty related to firn-model choice by running the suite of models across the entire range of the climates encountered on the ice sheets. There are additional sources of uncertainty in firn-model outputs beyond the model choice; for example, any full uncertainty analysis will also require consideration of uncertainties in the boundary conditions and how those propagate through the model. The CFM is well-suited for such an exercise.

This model application focuses on a single location; the fact that the models do not agree well at Summit does not necessarily

mean that they would not agree at other sites, but agreement is unlikely. If one model performs best at Summit, it does not necessarily indicate that that model is the "best firn model". Indeed, it may be the best model for Summit, but it may not work as well at other locations, and it is not obvious where one model might be "better" than another. A number of arguments could be made as to why it is inappropriate to compare these models, especially at a lone site in Greenland. For example, some of the models are potentially outdated and not in use any longer, certain models were tuned for a particular place and

may not be appropriately applied to Summit, and some models were intended for ice-core delta-age reconstruction rather than mass-balance corrections, or vice versa. Regardless, each of these models was at one point the state-of-the-art, and each was designed to simulate the same properties of the firn. If the model equations are a correct and complete representation of the physics governing firn evolution, a model should be able to simulate firn evolution accurately on all time scales and all spatial scales.

Our results demonstrate a need to improve our understanding of firn-densification physics, which may include both validation of existing models and development of new models. Unfortunately, data that are needed for the development of a purely physically-based model are still lacking; any model development in the near future will require a certain amount of empirical tuning. For example, a microstructure-based firn-densification model will need empirical parameterizations for evolution of



the microstructural properties. The addition of descriptions of physical processes such as grain growth to a model does not

necessarily result in a better model if those physics (and the initial and boundary conditions) are not well constrained. For example, *ART-T* includes grain growth, but it does not necessarily produce better results. Ultimately, research should be done to both (1) further our understanding of the microstructural evolution and underlying physics of firn evolution and (2) improve empirical models with observations of the firn's macroscale behavior.

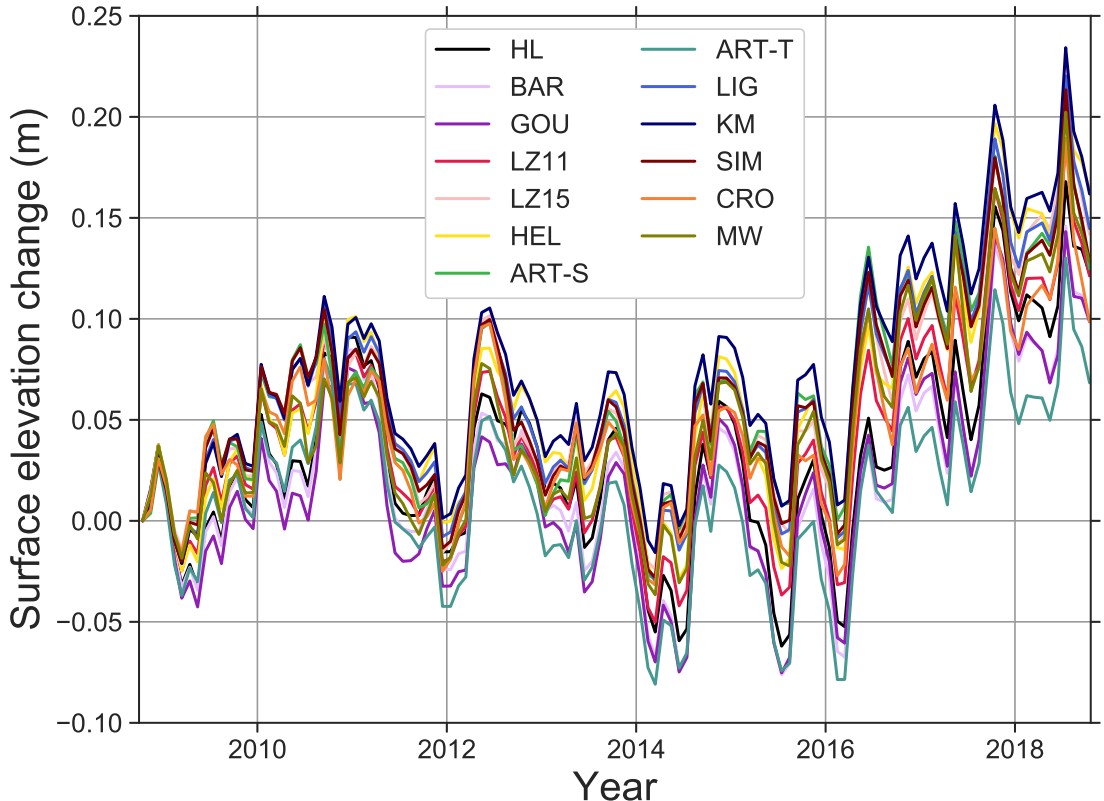

**Figure 2.** Model-predicted change in surface elevation at Summit, Greenland, from 2008 to 2018. This figure is also included in the supplementary material as a layered .pdf file.

## 4    Model Application 2: Firn-air stable isotopes and firn-thickness change during climate changes

In this application, we used the coupled firn-air and firn-densification models to investigate the effect that a thickening and thinning (i.e. non-steady-state) firn column has on nitrogen and argon isotope records in ice cores.

Gas isotopes fractionate in firn due to gravity (heavier isotopes become enriched at greater depths) and due to thermal gradients (heavier isotopes become enriched in the colder part of the firn column) (Severinghaus et al., 1998). In a steady





climate, there are seasonal temperature gradients near the surface (upper $\sim 10$m of firn), and the deeper firn is isothermal or near-isothermal due to the low temperature diffusivity of firn. During climate-change events, the surface warms or cools and creates a temperature gradient between the surface and the lock-in depth (LID), causing thermal fractionation. Different gas species have different thermal sensitivities, and this fact can be leveraged to infer the magnitude of temperature changes during rapid climate-change events. Previous work showed that the temperature at Summit increased by $\sim 9\,^\circ$C over the course of several decades during the Bølling Transition (14.67 ka before 1950; Severinghaus and Brook, 1999) and by $5 - 10\,^\circ$C over a century at the end of the Younger Dryas (11.6 ka before 1950; Severinghaus et al., 1998). In both these cases, the authors examined $\delta^{15}$N and $\delta^{40}$Ar isotopes; $\delta^{15}$N and $\delta^{40}$Ar/4 will have the same gravitational fractionation signal. Any deviation of these species from one another in the firn is a result of thermal fractionation. Those authors modeled gas isotopes to infer temperature increases, and part of their data-model mismatch was attributed to the fact that their model assumed a steady-state firn column. This model was unable to account for transient firn thickening due to an accumulation-rate increase coincident with the temperature increase; their model predicted values $\sim 3 - 4\,‰$ less than the observed values.

## 4.1 Methods

We used the CFM's coupled firn-air and firn-density models to examine the effect of transient firn evolution on gas records in ice cores during rapid climate-change events. We ran two model simulations of the evolution of $\delta^{15}$N and $\delta^{40}$Ar in the firn at Summit. We forced the CFM with temperature and accumulation-rate histories from the GISP2 ice core (Cuffey and Clow, 1997; Alley, 2000, 2004). Both simulations ran for 49,000 years, which is the length of the climate records. We ran the model using yearly time steps, and the model domain extended to a depth of $\sim 2200$m. The difference between the two model runs was the firn depth-density profile: in the first simulation, we used a constant profile; in the second, we used a transient firn-densification model to allow the density to evolve with the climate.

For the first simulation, we used a steady-state depth-density profile predicted by the Herron and Langway (1980) analytic model with an accumulation rate of $0.07\,$m ice eq. a$^{-1}$ and a temperature of $-47.5\,^\circ$C, which are consistent with values during the Younger Dryas and leading into the Bølling Transition and those used for modeling in Severinghaus et al. (1998) and Severinghaus and Brook (1999). In this simulation, we used the GISP2 temperature record (Fig. 3, upper panel) to force the CFM's temperature-evolution module, thereby allowing the firn temperature to evolve. However, we did not allow this temperature forcing to affect the model depth-density profile, and the LID stayed constant (Fig. 3, lower panel) at $\sim 97$m.

For the second simulation, we used the GISP2 temperature and accumulation-rate data (Fig. 3, upper panel) to run the CFM in transient mode using the Herron and Langway (1980) dynamic firn-densification model. In addition to the firn-temperature-profile evolution, this simulation allowed the firn depth-density profile to evolve, causing the LID to vary through time (Fig. 3, lower panel).

## 4.2 Firn-air results

Figure 4 shows the results of the two simulations and the $\delta^{15}$N and $\delta^{40}$Ar/4 data from Severinghaus et al. (1998) and Severinghaus and Brook (1999). The horizontal axis of the plot is the gas age. We add 1.5% to the modeled gas ages because the gas



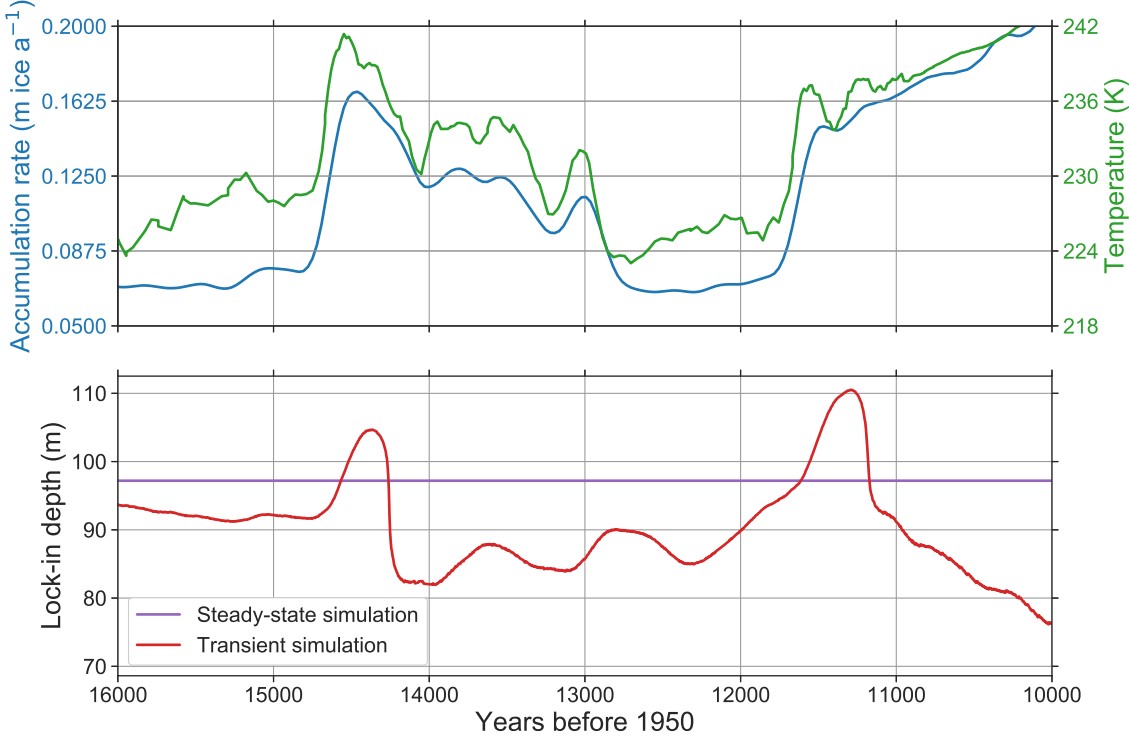

**Figure 3. Upper panel:** Accumulation-rate and temperature histories derived from the GISP2 ice core (Alley, 2004; Cuffey and Clow, 1997) and used to force the CFM for the analyses in Section 4. **Lower panel:** LID predicted by the steady-state and transient model simulations described in section 4. The LID for the transient simulation is determined by running the CFM with the climate shown in the upper panel, and the LID for the steady-state simulation is that predicted by the Herron and Langway (1980) model with accumulation rate $= 0.07\,\mathrm{m\,ice\,eq.\,a^{-1}}$ and mean annual temperature $-47.5\,^{\circ}\mathrm{C}$.

ages predicted by the model are too young compared to the data (determined by comparing the timing of the modeled isotope increases to the data during the Bølling Transition and Younger Dryas). The model is likely failing to produce the correct gas age for several reasons: (1) for our simple experiement, we assumed a uniform gas age of 15 years at the LID; (2) the large
modeled isotope changes occur when the firn-densification model, which is not necessarily accurate, predicts a firn thickness change; and (3) there are likely uncertainties in the climate forcing data, including with the timescale of those data. The $\delta^{15}\mathrm{N}$ and $\delta^{40}\mathrm{Ar}/4$ data were converted from depth to gas-age using the GICC05 timescale (Rasmussen et al., 2014; Seierstad et al., 2014).

For the steady-state simulations, variability in the isotope values is due only to fractionation from temperature gradients in
the firn. For the transient model runs, the variability is due to both fractionation from temperature gradients and to changes in the firn-column thickness. For the isotopes considered independently, much of the difference in the predicted isotope values between the transient and steady-state simulations shown in Fig. 4 can be attributed to the change in firn-column thickness. For



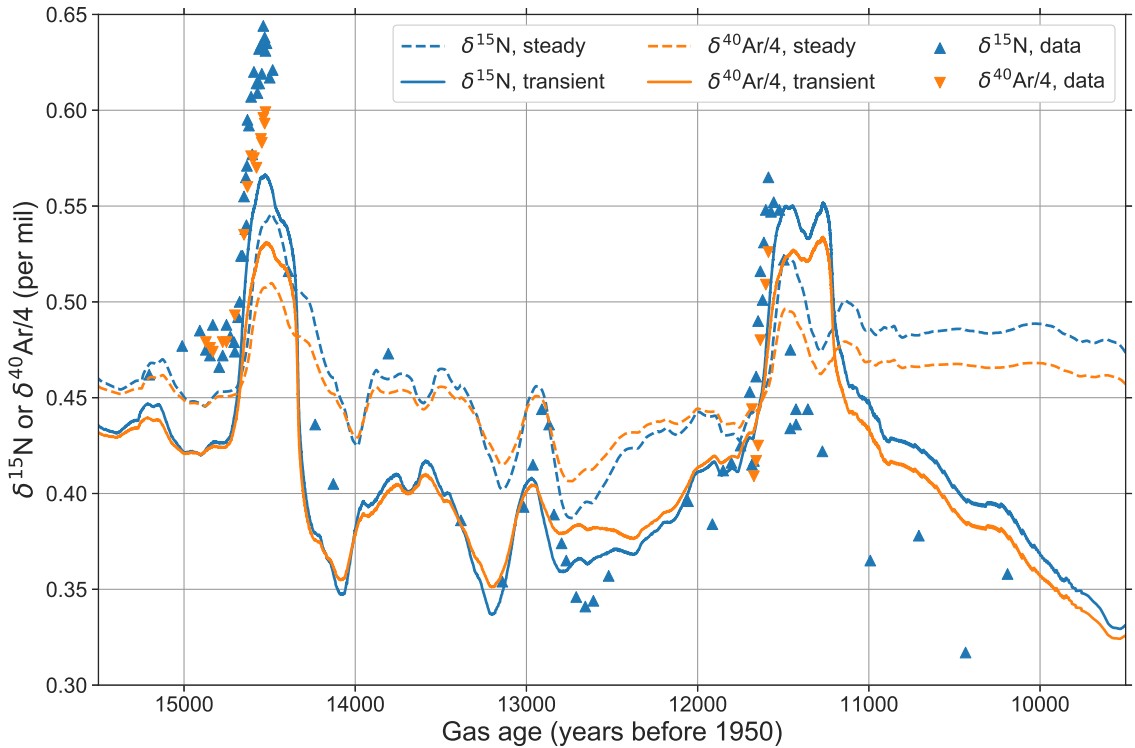

**Figure 4.** Measured and modeled $\delta^{15}$N and $\delta^{40}$Ar/4 profiles. Data are from Severinghaus et al. (1998) and Severinghaus and Brook (1999). We add 1.5% to the modeled gas ages to fit the data.

example, at $\sim 14{,}500$ years before present, the peak of the $\delta^{15}$N predicted by the transient model is about 0.02‰ higher than the steady-state run. In this case, the increased firn thickness results in more gravitational fractionation. However, the transient

and steady curves are also offset from one another temporally. This is because (1) the temperature gradients that form in the transient firn are different from the gradients in the steady-state firn; i.e. a thicker diffusive column in the firn will result in a different temperature gradient through time for the same surface temperature increase, and (2) the time scales of diffusion are slightly different; i.e. it will take longer for the thicker firn to come to a new thermal equilibrium.

On the whole, the transient model matches the data better than the steady-state model, especially during the Younger Dryas

and its termination. Notably, the transient model does well at predicting the high values of $\delta^{15}$N and $\delta^{40}$Ar/4 associated with the rapid warming at the end of the Younger Dryas. These values are $\sim 0.03$ ‰ higher than those predicted by the steady-state model and are consistent with the model-data misfit in Severinghaus et al. (1998), who used a fixed LID.

There is a $\sim 0.05$‰ offset between the data and transient model at $\sim 14.8$ka before present, suggesting that the model is predicting a LID that is $\sim 10$ m too shallow ($\sim 10\%$ of the firn column thickness) at the start of the Bølling Transition. This

may be caused by forcing-data uncertainty: around $-47.5\,^{\circ}$C and $0.07$ m ice eq. a$^{-1}$, the Herron and Langway (1980) model predicts a 4-m change in LID for a $1\,^{\circ}$C change in temperature and a 5-m change in LID for a $0.01$ m ice eq. a$^{-1}$ change in



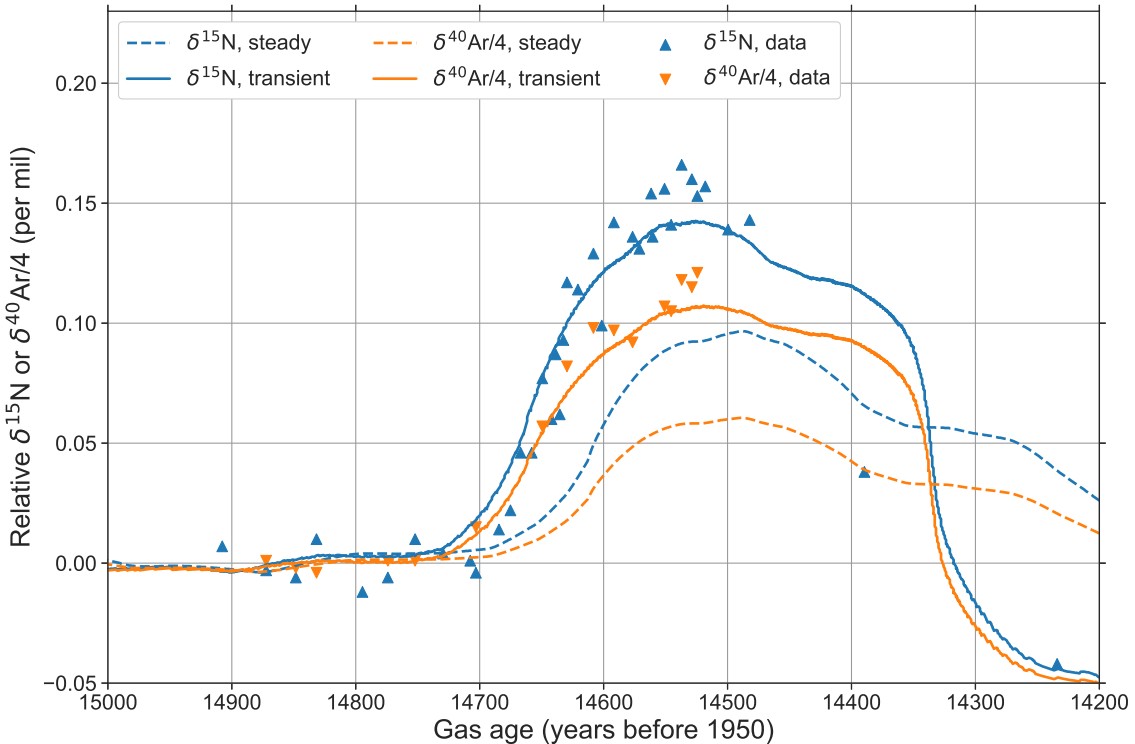

**Figure 5.** As in Fig. 4, but zoomed into the Bølling Transition. Additionally, for this figure we have shifted the y-axis to match the transient and steady-state simulations with the data leading into the Bølling Transition to highlight the magnitude of the modeled changes compared to the data.

accumulation rate; a small uncertainty in the forcing data for either of these variables could account for the 0.05 ‰ offset. The model-data discrepancy could also be caused by model inadequacy: the Herron and Langway (1980) model's calibration data set included only two cold, low accumulation sites, so its accuracy in those conditions may be questionable. The model also
may not be predicting a large enough temperature gradient, which could occur if the temperature increase in the forcing data was not large enough.

Figure 5 shows the data and model results zoomed in on the Bølling Transition. In order to directly compare the magnitude of the modeled $\delta^{15}$N and $\delta^{40}$Ar/4 to the data, we have shifted the data and the steady-state and transient isotope values by subtracting the mean $15 - 14.75$ ka B.P. $\delta^{15}$N values. The transient model matches the observed magnitude and rate of the
$\delta^{15}$N and $\delta^{40}$Ar/4 change much better than the steady-state model. This is because the transient CFM predicts that the LID increases from 92 m at 14.7 ka B.P. to 105 m at 14.4 ka B.P.

Despite the lingering model-data mismatch, the transient model clearly performs better than the steady-state model, and our model results during the Younger Dryas and Bølling Transition corroborate the assertion in Severinghaus and Brook (1999) that their model-data misfit is due to transient firn thickening. Our results do not change their conclusions but do provide





assurance that their conclusions are sound. This application also demonstrates the utility of the CFM's coupled firn-air/firn-density model. In this case, the coupled model should allow a more accurate assessment of the magnitude of temperatures because it can account for the temperature gradient that will form in thickening firn, which is smaller than that in steady-state firn. It is important to note that most firn-densification models (including Herron and Langway (1980)) were developed with a steady-state assumption, and applying them to transient simulations produces additional uncertainty. This uncertainty is

challenging to quantify, however, because we do not have direct observations of firn evolution during rapid climate changes.

The CFM's coupled firn-densification and firn-air modules have additional potential to help test hypotheses surrounding anomalies in ice core records. For example, experiments could be done to investigate the impact that an impermeable ice lens would have on gas records. It could also be used to model water-isotope diffusion simultaneously with firn-air transport. The CFM's ability to model multiple physical processes in a single framework allows us to investigate processes with different

timescales. For example, a temperature change (with no concurrent accumulation-rate change) will create a temperature gradient in the firn and will also cause the firn to change thickness by affecting the densification rate, but those processes will operate on different timescales.

## 5   Conclusions

We developed the Community Firn Model (CFM), an open-source firn-model framework. The CFM includes modules to

simulate a number of physical processes in firn, including densification, heat transport, meltwater percolation, grain growth, water-isotope diffusion, and firn-air diffusion. We demonstrated the utility of the CFM in two model applications. In the first, we leveraged the CFM's ability to run numerous firn-densification models by forcing 13 models with the regional climate model outputs for Summit, Greenland. These simulations showed that choice of firn-densification model can contribute significant uncertainty to firn-model outputs: the model spread was greater than 10% of the model mean for the metrics of depth-integrated

porosity, bubble close off depth and age, and surface-elevation change trend. There is no single densification model that is widely considered best; different models are preferred for different locations (e.g. Antarctica vs. Greenland) or for different applications (e.g. ice cores vs. satellite altimetry). Continued studies are necessary to improve firn-densification models and to better understand the uncertainty in their applications. These include investigations of the microstructural evolution of firn and in-situ measurements of the bulk firn-densification rate in a variety of climates.

In the second application, we investigated the effect of a thickening or thinning firn column on noble gas isotope records in ice cores. To our knowledge, the CFM is the first model that couples transient firn densification and firn-air transport. This application demonstrated that the coupled model can predict records of isotopes in ice cores better than a firn-air model that uses a steady-state firn density profile. This tool could be used for a number of studies surrounding firn air transport in changing climates. The model is limited in that it relies on published parameterizations of the effective diffusivity, which may fail to

incorporate all relevant parameters, especially near and in the lock-in zone. Continued research investigating microstructure at the bottom of the firn column is needed to improve our ability to model air transport accurately.

The goals of the CFM project are to provide a community resource that can be used by research groups that need a firn model and to provide the ability to create open-source results for model comparison and benchmarking. The CFM has already been used for several studies, including Verjans et al. (2019) and Garland et al. (2018). The CFM allows a fast and easy way to run
a model experiment using the same boundary conditions with different densification physics, and it removes potential sources of "noise" when comparing the outputs of different models, e.g. numerical solvers, different time stepping, etc. The code is open source, allowing anyone to be able to check results from researchers who use the CFM. As the firn-research community improves our understanding of physical processes in firn, e.g. new descriptions of densification or meltwater percolation processes, the new knowledge can be incorporated easily into the CFM's modular framework. We encourage other researchers to
develop their own modules and add them to the code.

*Code and data availability.* The CFM code is publicly available under the MIT license at https://github.com/UWGlaciology/CommunityFirnModel (Stevens et al., 2019). Its documentation is online at https://communityfirnmodel.readthedocs.io/. All model outputs and scripts used to make the figures are freely available upon request.

*Author contributions.* CMS was the main developer of the CFM, conceived and ran the model applications, and wrote the manuscript. EDW
initiated the CFM project and supervised its development. JL helped initiate the CFM project, planned model architecture, and developed code. VV, EK, AH, and BH developed code for the CFM. All authors contributed to manuscript writing and editing.

*Competing interests.* The authors declare that they have no conflict of interest.

*Acknowledgements.* This work was supported by U.S. National Science Foundation (NSF) grant 0968391 and National Aeronautics and Space Administration (NASA) grant NNX15AC62G. Thank you to P. Harris, A Le, W. Leahy, H. Vo, M. Yoon for assistance developing the
CFM. We used GNU Parallel (Tange, 2011) to parallelize model runs for application 1.



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
