# Peer review of "The Community Firn Model (CFM) v1.0"

_Geoscientific Model Development, 2019_

## Referee Comment (RC1) · Anonymous Referee #1 · 19 Mar 2020

This paper presents a new firn model (the Community Firn Model or CFM) which is open source (available on Github) and includes 13 previously published firn densification models. It also includes modules of firn densification, heat transport, meltwater percolation and refreezing, water isotope diffusion, and firn air diffusion. Users can easily choose model parameterizations by using different module options. They show two applications of this model including 1) forcing it with MAR3.9 SMB and skin temperature at Summit, Greenland to see a model comparison of firn densification and depth integrated porosity (DIP) and 2) the effect of thickening and thinning firn column on noble gas isotope records in ice cores.

This is an extremely important contribution to the firn modelling and ice sheet community. Having an open source, modular firn model with the framework already built in that is easily customizable is a huge step in the right direction of making our science more reproducible and advancing firn modelling. As a whole, the paper is well written

and generally straightforward.

Specific Comments:

1. Make sure that the manuscript is consistent in adding in reference observations (i.e. Table 1) and mentioning that this is only a model comparison study. As a reader, I don't mind the addition of observations for reference in the first model application; frankly I think it is helpful in understanding. However, it is heavily implied that there should be no takeaways of the "best" model and adding in these observations is a bit confusing to the overall message.

2. Both applications are done in a completely dry location (Summit, Greenland). There is a reference to using CFM to examine locations in the percolation zone in Verjans et al, 2019 and validate it using observations, but I would be curious to see how much more of a spread the models would have had if a location experiencing melt.

3. Since the CFM has already been used in some studies, it may be useful to outline how other published studies have already used the model in a few sentences, and not just reference them. I realize this is away from the main point of the paper (to describe the model), but it may be a good addition to the applications.

In-Line Comments:

L24 – "density" inside parentheses?

L27 – Mention zone 3 here.

L65-70 – Great job outlining clear goals of the paper in the last paragraph of the introduction.

L83 – Is there a default .json configuration file or do you have to choose all parameters individually? If there is not a default, are there recommended parameters for a first time user?

L83 – Are there limitations to the time step or the model-domain thickness?
L87 – Are these .csv files the forcing files? Normally MAR and RACMO come in netCDF files, do users need to reformat these prior to using?

L92 – Can you prescribe a number of layers or is it all based on volume? For example, if a large amount of accumulation was added and I want to look at a higher resolution vertically, could I break that accumulation into more layers?

L101 – How can you specify what resolution it is outputting at? Is that by specifying "firn depth"?

L159/L172 Be consistent in using the delta symbol or "delta".

L168 – Define as zone 3 equation.

Section 2.2.7 – Add in that LIG and KM included refreezing, and meltwater retention and percolation compared to Arthren.

L257 – radar layers from ground based or airborne instruments?

L330 – Include what Fick's second law is or a reference?

L344-346 – This sentence is difficult to follow. Maybe break into two?

L349 – How is the air moving upward relative to the downward moving grid? Because it is slower than the firn then the magnitude is upward because the downward moving grid is what it is referencing? Confused here, maybe rephrase.

L352 – In what scenarios would it be worth it to include this effect if previous models have ignored it?

L367 – Define gravitational enrichment.

L392 – Would there be a difference if looking at dendritic grains instead of spherical?

L443 – Reference for using 300 kgm-3 constant surface density at Summit?

L459 – How did you generate temperature and accumulation rate histories for 1000

years prior?

Table 1/L599-609 – There needs to be consistency between the text and this Table. The text continually repeats that this study is not to compare models to observations and rather to look at only model results. This is fine, but providing a reference is going to lead readers to make a direct comparison between the models and observations.

L512-520 – Here, the author compare the models to the observed core and provide reasons why they are different. As a reader, a takeaway is whether the models are doing "well" at that test site.

L573 – Unsure about using "trend" from 2008-2018 period. Isn't this most likely just inter-annual variability?

Figure 2 – Similar to my other figure comment, should a reference observation be provided here for dH/dt from Summit if a reference is provided for DIP and age earlier? Even if for just comparing to the model mean? Just make sure this is consistent.

L622 – Is this the gravitational enrichment definition?

L626 – I think LID is already defined from earlier.

—————————————

---

## Referee Comment (RC2) · Anonymous Referee #2 · 23 Mar 2020

Review of

The Community Firn Model (CFM) v1.0

by Stevens and others.

General

This is a laudable effort to synthesize efforts to model firn layers in a wide variety of climate conditions; the open-source model is expected to be widely used by remote sensing community (translating elevation change to mass change) but also by the regional climate modellers that aim at improving the representation of firn in their models, or simply as a standalone tool for process studies such as firn air age, pore close-off and the interpretation of firn gas records. The paper is well structured, clearly and concisely written and the figures of good quality. My comments are relatively minor.

General comments

l.87: Perhaps this is a good point to state specifically whether shortwave radiation penetration has been included (no), with a short motivation why (not).

l. 88: H-L is an explicit expression that directly produces a profile without the necessity of time stepping? Furthermore, assuming constant accumulation rate and surface temperature to spin up the model is not very realistic and will lead to a 'shock' once the real forcing data are applied. Is there an option to simply repeat the first (couple of) year(s) to spin up the model, to reduce this shock? This is done later in the application sections, but it would be worthwhile to already state that possibility here.

l. 440: Monthly time steps appear long, and ignore the impact of the daily cycle and the a-synchronicity of snowfall and temperature fluctuations; has it been tested how different the results look if daily or hourly time steps are taken?

l.541: Inferences are made about the possible reasons for the mismatch between the models and the observations. This ignores that fact that the forcing model data also are uncertain and responsible for part of the mismatch. It would be interesting to briefly explore this sensitivity (varying temperature by +/- 1 degree and accumulation by +/- 10%).

Minor comments l. 24: "The first zone is defined to include the firn from the surface density". This is unclear.

l. 105: models -> expressions (?) OK, explained later on, perhaps start with the explanation.

l.386: "the firn temperature is raised by an amount that equals the latent heat released by refreezing" Suggest to replace "equals" with "corresponds to".

l. 508: "The firn-core density data have high variability with depth" What causes these variations, surely not melt?

---

## Author Comment (AC1) · 20 Apr 2020

**We thank both referees for their questions and comments; the changes they elicited strengthened the paper. We have addressed each of the questions and comments below. The most notable change we made, based on the comments from both of the reviewers, was to remove the firn-core data, which confused our message while not providing significant substance.**

**Reviewer #1 Comments**

This paper presents a new firn model (the Community Firn Model or CFM) which is open source (available on Github) and includes 13 previously published firn densification models. It also includes modules of firn densification, heat transport, meltwater percolation and refreezing, water isotope diffusion, and firn air diffusion. Users can easily choose model parameterizations by using different module options. They show two applications of this model including 1) forcing it with MAR3.9 SMB and skin temperature at Summit, Greenland to see a model comparison of firn densification and depth integrated porosity (DIP) and 2) the effect of thickening and thinning firn column on noble gas isotope records in ice cores.

This is an extremely important contribution to the firn modelling and ice sheet community. Having an open source, modular firn model with the framework already built in that is easily customizable is a huge step in the right direction of making our science more reproducible and advancing firn modelling. As a whole, the paper is well written and generally straightforward.

Specific Comments:

1. Make sure that the manuscript is consistent in adding in reference observations (i.e. Table 1) and mentioning that this is only a model comparison study. As a reader, I don't mind the addition of observations for reference in the first model application; frankly I think it is helpful in understanding. However, it is heavily implied that there should be no takeaways of the "best" model and adding in these observations is a bit confusing to the overall message.

Thanks for this observation. When writing the paper, we debated whether or not to include the core data. We decided to remove the data from the depth-density and depth-DIP plots, but we kept it in the table for reference, also thinking it was helpful. But, based on comments from both reviewers, we have decided to remove the core data from the study and focus solely on the model intercomparison.

2. Both applications are done in a completely dry location (Summit, Greenland). There is a reference to using CFM to examine locations in the percolation zone in Verjans et al, 2019 and validate it using observations, but I would be curious to see how much more of a spread the models would have had if a location experiencing melt.

We agree that this would be an interesting addition to this study, and we actually considered including such a comparison while we were preparing the manuscript. However, we opted to keep the present study limited to a simple model comparison at a dry firn site, because our primary goal was to provide a model description paper. The several examples were designed to demonstrate the CFM's utility but were not meant to be an exhaustive study of firn-model uncertainty. We do have additional model studies in the pipeline, including a model comparison at numerous wet-firn sites in Greenland.

The short answer to the reviewer's question is that there is much more spread. Verjans et al. (2019) compared HL, KM and Cr at four sites with different melting rates. Verjans et al. (2019) is more focused on comparing meltwater schemes, but still shows that the different densification models cause large spread in model results.

Additionally, since submission of the present paper, the Firn meltwater Retention Model Intercomparison Project (RetMIP) has been submitted to the Cryosphere, and it includes results from the CFM (as well as numerous other coupled firn densification/hydrology models.)

3. Since the CFM has already been used in some studies, it may be useful to outline how other published studies have already used the model in a few sentences, and not just reference them. I realize this is away from the main point of the paper (to describe the model), but it may be a good addition to the applications.

This is a good suggestion. We have added a short section to describe those studies.

In-Line Comments:

L24 – "density" inside parentheses?

We agree that that sentence was unclear. Changed to: "The first zone extends from the surface, where density is often assumed to be ~300-350kg m$_{-3}$, to 550 kg m$_{-3}$."

L27 – Mention zone 3 here.

Done. We slightly removed the veiled reference to zone 3 and added the following: "Further densification occurs due to compression of the bubbles in zone three, which comprises the firn/bubbly ice between the BCO density and the ice density."

L65-70 – Great job outlining clear goals of the paper in the last paragraph of the introduction.

Thank you.

L83 – Is there a default .json configuration file or do you have to choose all parameters individually? If there is not a default, are there recommended parameters for a first time user?

There is an example .json configuration file on the GitHub repository. We added the following sentence to clarify: "The CFM's GitHub repository includes an example configuration file preset with default values, and the CFM's documentation includes detailed descriptions of each of the parameters."

L83 – Are there limitations to the time step or the model-domain thickness?

There are no limits on these parameters, though the computing time increases with a thicker domain and/or shorter time steps. We have restructured the text slightly and added several sentences to clarify and explain this.

L87 – Are these .csv files the forcing files? Normally MAR and RACMO come in netCDF files, do users need to reformat these prior to using?

Yes, they need to be reformatted. We recognize that this may be is a hurdle for some users. However, creating a flexible script to accommodate different inputs is actually quite challenging because although climate data are commonly in .netCDF files, the way the data is organized in those files is not standardized. For example, some regional climate products use a time stamp of 'days since' and others use 'months since'. The RACMO data that we have worked with has been provided as individual files for each field, spanning a number of years (e.g. skin temperature, 2011-2018), where MAR comes as all fields (temperature, precip, melt, etc.) for a single year in one file. Ice core data often is provided in text files with a time stamp of years before present, but the temperature and accumulation records often have

their data at different times (e.g. there is a temperature given at 34.45 ka bp, but accumulation is given at 34.42 ka bp).

We wrote additional text to clarify this. We also wrote a script and uploaded it to the CFM's GitHub repository that provides guidance on how to convert the climate fields from the .netCDF files into .csv files appropriate for the CFM.

L92 – Can you prescribe a number of layers or is it all based on volume? For example, if a large amount of accumulation was added and I want to look at a higher resolution vertically, could I break that accumulation into more layers?

It is all based on volume, because a new layer is added at each time step, which is that time step's accumulation. That layer is assumed to have homogenous properties. We edited the text to clarify that this is the case, and we added the sentence, 'The number of volumes is determined by the thickness of the model domain, the time step size, and the mean-annual accumulation rate." In theory, we could modify the code to divide layers, though in our experience there is more of a need to combine layers/reduce layers to reduce model run time.

We are eager to improve the CFM based on user needs, so if a user requested the ability to split nodes we would be happy to try to include that in a future release. We changed the final sentence slightly to indicate our willingness to incorporate new features: "In the spirit of open-source software, we encourage other researchers to develop their own modules and add them to the code or to request features that would improve the CFM's usefulness and/or ease of use."

L101 – How can you specify what resolution it is outputting at? Is that by specifying "firn depth"?

The model by default saves the outputs on the entire grid, but that can be altered in the configuration file. We added the text, "The resolution of the model outputs is specified by the user in the .json configuration file; by default the CFM saves the outputs on the entire model grid at each time step."

L159/L172 Be consistent in using the delta symbol or "delta".

Fixed to use delta symbol consistently.

L168 – Define as zone 3 equation.

Specified Zone 3

Section 2.2.7 – Add in that LIG and KM included refreezing, and meltwater retention and percolation compared to Arthren.

We respectfully disagree that this is an important distinction in this section, because we are focusing on the densification equations. Our interpretation of LIG and KM is that the authors used depth-density profiles from the ice sheets to modify the Arthern equations. Although a number of the profiles may be from wet-firn sites, the densification equations for LIG and KM do not include melt (they are only dependent on accumulation rate, temperature, and density).

We did restructure the paragraph to give a bit more information about KM and LIG by indicating that they comprise the RACMO subsurface scheme, which includes meltwater percolation and refreezing. We also changed the language from 'LIG and KM are the same as ART-S with the exception' to 'LIG and KM use the same form as ART-S with the exception.'

L257 – radar layers from ground based or airborne instruments?

Airborne; now specified

L330 – Include what Fick's second law is or a reference?

Added Fick's law equation and expanded description of the isotope diffusion process.

L344-346 – This sentence is difficult to follow. Maybe break into two?

Agree this was poorly worded; we split into 3 sentences

L349 – How is the air moving upward relative to the downward moving grid? Because it is slower than the firn then the magnitude is upward because the downward moving grid is what it is referencing? Confused here, maybe rephrase.

Agree this was not written clearly; we removed the reference to upward moving air and clarified the text about the pressure gradient. The important point is that the air advection rate is slower than the firn advection rate.

L352 – In what scenarios would it be worth it to include this effect if previous models have ignored it?

Not all previous firn-air models have ignored it; the Trudinger and others (1997) model is the only other Lagrangian firn-air model we know of. We clarified the text to indicate that it is included in some Eulerian firn air models. Our goal is to maximize the flexibility of the CFM, so we include the option to include/exclude this feature.

L367 – Define gravitational enrichment.

Changed wording to "gravitational fractionation", which is defined in the introduction.

L392 – Would there be a difference if looking at dendritic grains instead of spherical?

Probably – the parameterizations we include in the CFM are taken from other papers, so we fall back on the assumptions from those papers. We changed the wording of this sentence to indicate that we are following the assumptions of those authors.

We also changed the title of the section to "Grain growth and microstructure evolution" to better indicate that there are numerous microstructure properties that we would like to include in future releases. We changed the final paragraph of this section to indicate that the CFM's microstructure module is a work in progress:

"We recognize that many macroscale processes in firn (e.g. bulk densification) are dependent on the firn's microstructure (e.g. grain shape, size, and coordination number; specific surface area). Unfortunately, at present there is a lack of research describing evolution of polar firn microstructure and how microstructure relates to macroscale firn processes. We have strived to design the CFM so that equations describing microscale evolution can be easily integrated into its framework. We will incorporate these equations when future research provides insights into how those properties evolve."

L443 – Reference for using 300 kgm-3 constant surface density at Summit?

Reference to the SUMup dataset (Montgomery and others, 2018) added.

L459 – How did you generate temperature and accumulation rate histories for 1000 years prior?

We repeated the 1958 to 1978 climate over and over again – this was mentioned later in the paragraph, but we restructured the paragraph to clarify.

Table 1/L599-609 – There needs to be consistency between the text and this Table. The text continually repeats that this study is not to compare models to observations and rather to look at only model results. This is fine, but providing a reference is going to lead readers to make a direct comparison between the models and observations.

We removed the core data from the table and text.

L512-520 – Here, the author compare the models to the observed core and provide reasons why they are different. As a reader, a takeaway is whether the models are doing "well" at that test site.

Removed core data to focus on the model intercomparison.

L573 – Unsure about using "trend" from 2008-2018 period. Isn't this most likely just inter-annual variability?

The 'trend' may be inter-annual variability, but we feel that it is a useful metric to inter-compare the models. It allows us a quantitative look at how the models respond differently to the inter-annual climate variability. All of the models predict that the surface elevation increased from 2008 to 2018. An alternative method could have been to look at just the final dH value in our time series (i.e. in October 2018), but we feel the "trend" provides a better comparison metric for the model spread. The take-home message is that when modeling firn to simulate surface elevation change, the model choice can change the answer significantly, which we discuss in the next paragraph.

Figure 2 – Similar to my other figure comment, should a reference observation be provided here for dH/dt from Summit if a reference is provided for DIP and age earlier? Even if for just comparing to the model mean? Just make sure this is consistent.

Removed observations for DIP and age section.

L622 – Is this the gravitational enrichment definition? L626 – I think LID is already defined from earlier.

Yes – see response to comment about L367. LID was indeed previously defined. When preparing the manuscript we decided it would be prudent to include the definition again in this new section that is many pages later. We feel that in the case that the reader is unfamiliar with the LID, he or she may appreciate a reminder of what it is, and the reader familiar with the concept is unlikely to be put off by its inclusion again. We would not be opposed to its removal here if the referee or editor deems it necessary, but we believe it adds clarity for the reader.
* * *
**Reviewer #2**
**General**
This is a laudable effort to synthesize efforts to model firn layers in a wide variety of climate conditions; the open-source model is expected to be widely used by remote sensing community (translating elevation

change to mass change) but also by the regional climate modellers that aim at improving the representation of firn in their models, or simply as a standalone tool for process studies such as firn air age, pore close-off and the interpretation of firn gas records. The paper is well structured, clearly and concisely written and the figures of good quality. My comments are relatively minor.

**General comments**

l.87: Perhaps this is a good point to state specifically whether shortwave radiation penetration has been included (no), with a short motivation why (not).

The reviewer is correct; it is not included. We agree it is worthy to note that this is the case, but we think the appropriate place to add this information is in the "temperature evolution" section (2.3). We added the following paragraph:

"The CFM does not incorporate a scheme to account for the impact of shortwave radiation penetration into the firn, although research has suggested that it can affect the temperature in the near-surface snow by several degrees \citep{Kuipers2009}. Adding a module to account for this effect could be an area for future development of the CFM. This would require the implementation of a scheme that computes the transfer of these radiative components into firn, forced by surface values that must be provided by Regional Climate Models or weather station data."

l. 88: H-L is an explicit expression that directly produces a profile without the necessity of time stepping? Furthermore, assuming constant accumulation rate and surface temperature to spin up the model is not very realistic and will lead to a 'shock' once the real forcing data are applied. Is there an option to simply repeat the first (couple of) year(s) to spin up the model, to reduce this shock? This is done later in the application sections, but it would be worthwhile to already state that possibility here.

Herron and Langway (1980) present several sets of equations, including analytic equations that predict steady-state depth/density and depth/age profiles given mean annual temperature and accumulation rate. We use these to produce a guess of the depth-density profile - the model just needs some semi-realistic depth-density profile that can evolve during spin up to the initial condition.

The reviewer is correct that the model would receive a shock if it went straight from the H&L guess to the main model run. That does not ever occur (unless the user does not specify a long enough spin up time), and we recognize that we did a poor job explaining how the spin up process works. We rewrote the spin up paragraph to clarify.

l. 440: Monthly time steps appear long, and ignore the impact of the daily cycle and the a-synchronicity of snowfall and temperature fluctuations; has it been tested how different the results look if daily or hourly time steps are taken?

We have not tested with hourly time steps, but we have with daily time steps. The absolute numbers predicted by the models are different, but the model spread is similar. We feel that results with monthly time steps are adequate to demonstrate the spread in the models' outputs when forced by the same inputs. We think the results demonstrate that an investigation looking at how the time-step size affects the results is worthy of its own study, and the CFM would be well-suited to this task.

l.541: Inferences are made about the possible reasons for the mismatch between the models and the observations. This ignores that fact that the forcing model data also are uncertain and responsible for part of the mismatch. It would be interesting to briefly explore this sensitivity (varying temperature by +/- 1 degree and accumulation by +/- 10%).

This is a very good point, and upon consideration of it, along with comments from the other reviewer, we have decided to take out the comparison to the firn core data.

The reviewer is correct that the forcing data can be a significant source of uncertainty in the firn model outputs. We believe that this is a topic worthy of its own study (we mention this in the paragraph beginning at line 428). Including that study in the current manuscript would veer significantly from our primary goal of providing a model-description paper.

**Minor comments**
l. 24: "The first zone is defined to include the firn from the surface density". This is unclear.

We agree that that sentence was unclear. Changed to: "The first zone extends from the surface, where density is often assumed to be ~300-350kg m$_{-3}$, to 550 kg m$_{-3}$."

l. 105: models -> expressions (?) OK, explained later on, perhaps start with the explanation.

We changed the paragraph starting at L105 slightly to clarify this.

l.386: "the firn temperature is raised by an amount that equals the latent heat released by refreezing" Suggest to replace "equals" with "corresponds to".

Good suggestion; changed

l. 508: "The firn-core density data have high variability with depth" What causes these variations, surely not melt?

Upon consideration after reading both reviewers' comments, we have removed the firn-core data from the present study, so this observation is no longer in the paper.

The answer to the reviewer's question: this phenomenon has been widely observed (e.g. see Hörhold and others, JGR, 2011, https://doi.org/10.1029/2009JF001630 and references therein), but to our knowledge the exact cause of the high variability, especially at depth, is still debated. Hypotheses include that it is the result of impurities in the firn (e.g. Freitag and others, J. Glac, 2017, https://doi.org/10.3189/2013JoG13J042) that affect the densification rate or that it results from grain-scale heterogeneity at deposition that perseveres as the firn ages
(e.g. Morris and Wingham, JGR, 2014 https://doi.org/10.1002/2013JF002898).